# SCALABLE AND PRIVACY-ENHANCED GRAPH GENERATIVE MODEL FOR GRAPH NEURAL NETWORKS

## ABSTRACT

As the field of Graph Neural Networks (GNN) continues to grow, it experiences a corresponding increase in the need for large, real-world datasets to train and test new GNN models on challenging, realistic problems. Unfortunately, such graph datasets are often generated from online, highly privacy-restricted ecosystems, which makes research and development on these datasets hard, if not impossible. This greatly reduces the amount of benchmark graphs available to researchers, causing the field to rely only on a handful of publicly-available datasets. To address this dilemma, we introduce a novel graph generative model, Computation Graph Transformer (CGT) that can learn and reproduce the distribution of real-world graphs in a privacy-enhanced way. Our proposed model (1) generates effective benchmark graphs on which GNNs show similar task performance as on the source graphs, (2) scales to process large-scale real-world graphs, (3) guarantees privacy for end-users. Extensive experiments across a vast body of graph generative models show that only our model can successfully generate privacy-controlled, synthetic substitutes of large-scale real-world graphs that can be effectively used to evaluate GNN models.

## 1 INTRODUCTION

Graph Neural Networks (GNNs) (Kipf & Welling, 2016a; Chami et al., 2022) are machine learning models that learn the dependences in graphs via message passing between nodes. Various GNN models have been widely applied on a variety of industrial domains such as misinformation detection (Benamira et al., 2019), financial fraud detection (Wang et al., 2019a), traffic prediction (Zhao et al., 2019), and social recommendation (Ying et al., 2018). However, datasets from these industrial tasks are overwhelmingly proprietary and privacy-restricted and thus almost always unavailable for researchers to study or evaluate new GNN architectures. This state-of-affairs means that in many cases, GNN models cannot be trained or evaluated on graphs that are appropriate for the actual tasks that they need to execute. This scarcity of real-world benchmark graphs also leaves GNN researchers with only a handful of public datasets, which could potentially cause new GNN architectures to optimize performance only on these public datasets rather than generalizing (Palowitch et al., 2022).

In this paper, we introduce a novel graph generative model to overcome the unavailability of critical real-world graph datasets. While there is already a vast body of work on graph generation (You et al., 2018; Liao et al., 2019; Simonovsky & Komodakis, 2018; Grover et al., 2019), including differentially-private generation (Qin et al., 2017; Proserpio et al., 2012), we found that no one study has addressed all aspects of the modern GNN problem setting, such as handling large-scale graphs and node attributes/labels. We thus propose a novel, *modern* graph generation problem definition:

**Problem Definition 1.** *Let $\mathcal{A}$, $\mathcal{X}$, and $\mathcal{Y}$ denote adjacency, node attribute, and node label matrices; given an original graph $\mathcal{G} = (\mathcal{A}, \mathcal{X}, \mathcal{Y})$, generate a synthetic graph $\mathcal{G}' = (\mathcal{A}', \mathcal{X}', \mathcal{Y}')$ satisfying:*

- ***Benchmark effectiveness:*** *performance rankings among $m$ GNN models on $\mathcal{G}'$ should be similar to the rankings among the same $m$ GNN models on $\mathcal{G}$.*
- ***Scalability:*** *computation complexity of graph generation should be linearly proportional to the size of the original graph $O(|\mathcal{G}|)$ (e.g., number of nodes or edges).*
- ***Privacy guarantee:*** *any syntactic privacy notions are given to end users (e.g., k-anonymity).*

To address this problem statement, we introduce the Computation Graph Transformer (CGT) as the core of a graph generation approach with two novel components. First, CGT operates on *minibatches* rather than the whole graph, avoiding scalability issues encountered with nearly all existing graph generative models. Note that each minibatch is in fact a GNN *computation graph* (Hamilton et al., 2017) having its own adjacency and feature submatrices, and the set of all minibatches comprises a graph minibatch distribution that can be learned by an appropriate generative model.

Second, instead of attempting to learn the joint distribution of adjacency matrices and feature matrices, we derive a novel *duplicate encoding* scheme that transforms a $(A, X)$ adjacency and feature matrix pair into a single, dense feature matrix that is isomorphic to the original pair. In this way we are able to reduce the task of learning graph distributions to learning feature vector sequence distributions, which we approach with a novel Transformer architecture (Vaswani et al., 2017). This reduction is the key innovation allowing CGT to be an effective generator of realistic datasets for GNN research. In addition, after the reduction process, our model can be easily extended to provide $k$-anonymity or differential privacy guarantees on node attributes and edge distributions.

To show the effectiveness of CGT, we design three experiments that examine its scalability, its benchmark effectiveness as a substitute generator of source graphs, and its privacy-performance trade-off. Specifically, to examine this benchmark aspect, we perturb various aspects of the GNN models and datasets, and check that these perturbations bring the same empirical effect on GNN performance on both the original and generated graphs. In total, our contributions are: 1) we propose a novel graph generation problem featuring three requirements in state-of-the-art graph learning settings; 2) we reframe the problem of learning a distribution of a whole graph into learning the distribution of minibatches that are consumed by GNN models; 3) we propose the Computation Graph Transformer, an architecture that casts the problem of computation graph generation as conditional sequence modeling; and finally 4) we show that the test performance of 9 GNN models in 14 different task scenarios is consistent across 7 real-world graphs and their corresponding synthetic graphs.

## 2 RELATED WORK

**Traditional graph generative models** extract common patterns among real-world graphs (e.g. nodes/edge/triangle counts, degree distribution, graph diameter, clustering coefficient) (Chakrabarti & Faloutsos, 2006) and generate synthetic graphs following a few heuristic rules (Erdős et al., 1960; Leskovec et al., 2010; Leskovec & Faloutsos, 2007; Albert & Barabási, 2002). However, they cannot generate unseen patterns on synthetic graphs (You et al., 2018). More importantly, most of them generate only graph structures, sometimes with low-dimensional boolean node attributes (Eswaran et al., 2018). **General-purpose deep graph generative models** exploit GAN (Goodfellow et al., 2014), VAE (Kingma & Welling, 2013), and RNN (Zaremba et al., 2014) to learn graph distributions (Guo & Zhao, 2020). Most of them focus on learning graph structures (You et al., 2018; Liao et al., 2019; Simonovsky & Komodakis, 2018; Grover et al., 2019), thus their evaluation metrics are graph statistics such as orbit counts, degree coefficients, and clustering coefficients which do not consider quality of generated node attributes and labels. **Molecule graph generative models** are actively studied for generating promising candidate molecules using VAE (Jin et al., 2018), GAN (De Cao & Kipf, 2018), RNN (Popova et al., 2019), and recently invertible flow models (Shi et al., 2020; Luo et al., 2021). However, most of their architectures are specialized to small-scaled molecule graphs (e.g., 38 nodes per graph in the ZINC datasets) with low-dimensional attribute space (e.g., 9 boolean node attributes indicating atom types) and distinct molecule-related information (e.g., SMILES representation or chemical structures such as bonds and rings) (Suhail et al., 2021).

## 3 FROM GRAPH GENERATION TO SEQUENCE GENERATION

To develop a scalable and privacy-enhanced benchmark graph generative model for GNNs, we first look into how GNNs process a given graph $\mathcal{G}$. With $n$ nodes and $d$-dimensional node attribute vectors, $\mathcal{G}$ is commonly given as a triad of adjacency matrix $\mathcal{A} \in \mathbb{R}^{n \times n}$, node attribute matrix $\mathcal{X} \in \mathbb{R}^{n \times d}$, and node label matrix $\mathcal{Y} \in \mathbb{R}^n$. In this section, we illustrate how to convert the whole-graph generation problem into a discrete-valued sequence generation problem.

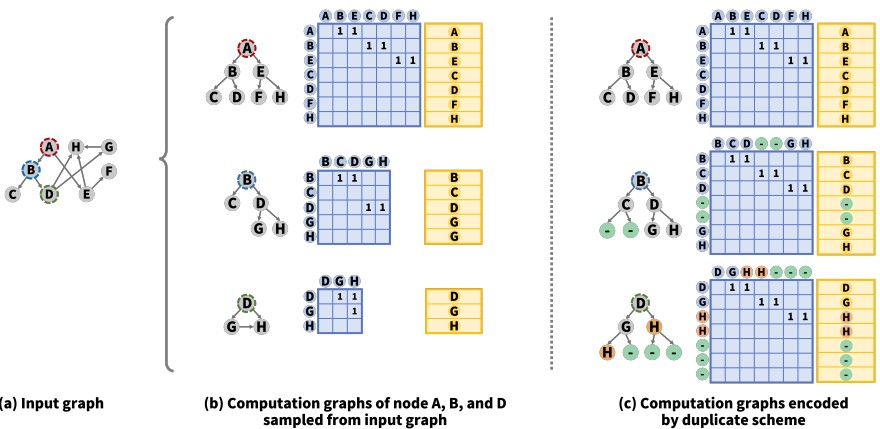

| (a) Input graph | (b) Computation graphs of node A, B, and D sampled from input graph | (c) Computation graphs encoded by duplicate scheme |

Figure 1: 2-**layered computation graphs with** $s = 2$ **neighbor samples:** (a) input graph; (b) original encoding scheme results in differently-shaped adjacency (blue) and attribute (yellow) matrices per computation graph; (c) duplicate encoding scheme outputs the *same adjacency matrix* and *identically-shaped* attribute matrices.

## 3.1 COMPUTATION GRAPHS IN MINIBATCH-BASED GNN TRAINING

To compute embeddings of node $v$, $L$-layered GNNs extract the node's $L$-hop egonet $\mathcal{G}_v$, namely the *computation graph*. Specifically, as with the global graph, $\mathcal{G}_v$ is composed of a sub-adjacency matrix $\mathcal{A}_v \in \mathbb{R}^{n_v \times n_v}$, a sub-feature matrix $\mathcal{X}_v \in \mathbb{R}^{n_v \times d}$, and node $v$'s label $\mathcal{Y}_v \in \mathbb{R}$, where each of $n_v$ rows correspond to nodes sampled into the computation graph. Minibatch-based GNN training samples one computation graph per each node in minibatch and runs GNN models on those computation graphs which are much smaller than the whole graph. Based on this observation, our problem reduces to: *given a set of computation graphs $\{\mathcal{G}_v = (\mathcal{A}_v, \mathcal{X}_v, \mathcal{Y}_v) : v \in \mathcal{G}\}$ sampled from an original graph, we generate a set of computation graphs $\{\mathcal{G}'_v = (\mathcal{A}'_v, \mathcal{X}'_v, \mathcal{Y}'_v)\}$.* This reframing shares intuition with mini-batch stochastic gradient descent that the distribution of randomly chosen subsets approximates the distribution of the original set (Bottou, 2010).

## 3.2 ENCODING SCHEME FOR COMPUTATION GRAPHS

In this work, we sample a fixed-size set of neighbors to generate computation graphs instead of using the full neighborhood, as proposed by GraphSage (Hamilton et al., 2017), a technique also widely adopted in popular GNN libraries (Fey & Lenssen, 2019; Ferludin et al., 2022; Wang et al., 2019b) to fix the minibatch computational footprint. To train a $L$-layered GNN model with a user-specified neighbor sampling number $s$, a computation graph is generated for each node in a top-down manner ($l : L \to 1$): A target node $v$ is located at the $L$-th layer; the target node samples $s$ neighbors, and the sampled $s$ nodes are located at the $(L - 1)$-th layer; each node samples $s$ neighbors, and the sampled $s^2$ nodes are located at the $(L - 2)$-th layer; repeat until the 1-st layer. When the neighborhood is smaller than $s$, we sample all existing neighbors of the node.

Generating a computation graph is similar to generating a balanced $s$-nary tree structure. For example, a balanced binary tree-shaped computation graph is generated for node $A$ in Figure 1(b) with neighbor sampling number $s = 2$. However, in practice, computation graphs are almost always unbalanced $s$-nary trees due to one of two cases: (1) lack of neighbors, and (2) neighbor sharing. In Figure 1(b), $B$'s computation graph is an unbalanced tree because node $C$ has no neighbors (case 1). In $D$'s computation graph, nodes $D$ and $G$ share node $H$ as neighbors, creating a cycle in the computation graph (case 2). These two cases result in variably-shaped of adjacency and node attribute matrices of computation graphs shown as blue and yellow boxes in Figure 1 (b).

## 3.3 DUPLICATE ENCODING SCHEME FOR COMPUTATION GRAPHS

We introduce a duplicate encoding scheme for computation graphs that is conceptually simple but brings a significant consequence: it *fixes the adjacency matrix for all computation graphs*, allowing us to model it as a constant. To circumvent case 1 from the previous paragraph, the duplicate encoding scheme defines a null node with zero attribute vector (node '−' in Figure 1(c)) and samples it as a padding neighbor for any node with less than $s$ neighbors. To circumvent case 2, the duplicate

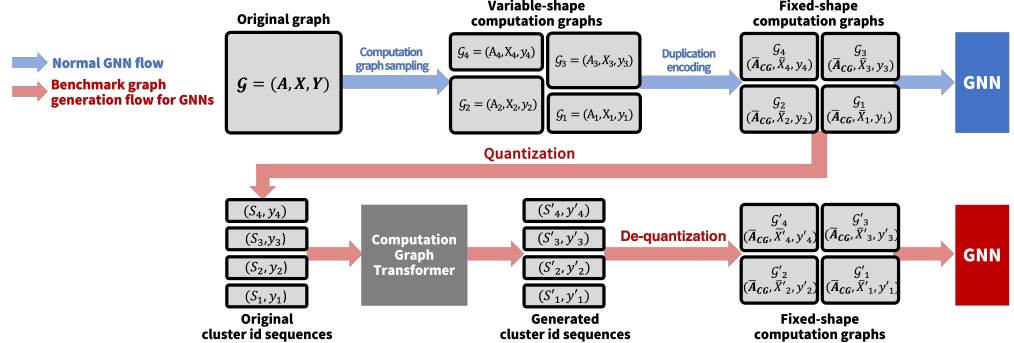

Figure 2: **Overview of our benchmark graph generation framework:** (1) We sample a set of computation graphs of variable shapes from the original graph, then (2) duplicate-encode them to fix adjacency matrices to a constant. (3) Duplicate-encoded feature matrices are quantized into cluster id sequences and fed into our Computation Graph Transformer. (4) Generated cluster id sequences are de-quantized back into duplicate-encoded feature matrices and fed into GNN models with the constant adjacency matrix.

encoding scheme copies shared neighbors and provides each copy to parent nodes (node $H$ in node $D$'s computation graph is copied in Figure 1(c)). Each node attribute vector is also copied and added to the feature matrix. As shown in Figure 1(c), the duplicate encoding scheme ensures that all computation graphs have an identical adjacency matrix (presenting a balanced $s$-nary tree) and an identical shape of feature matrices. Note that in order to fix the adjacency matrix, we need to fix the order of nodes in adjacency and attribute matrices (e.g., breadth-first ordering in Figure 1(c)).

Because our duplicate encoding scheme fixes the adjacency structure over all computation graphs, our problem reduces to learning the distribution of (duplicate-encoded) feature matrices of computation graphs, formalized as: *given a set of feature matrix-label pairs $\{(\tilde{\mathcal{X}}_v, \mathcal{Y}_v) : v \in \mathcal{G}\}$ of duplicate-encoded computation graphs, we generate a set of feature matrix-label pairs $\{(\tilde{\mathcal{X}}'_v, \mathcal{Y}'_v)\}$.*

## 3.4 QUANTIZATION

To learn distributions of feature matrices of computation graphs, we first quantize feature vectors into discrete bins; specifically, we cluster feature vectors in the original graph using k-means and map each feature vector to its (discrete) cluster id. Quantization is motivated by 1) privacy benefits and 2) ease of modeling. By mapping different feature vectors (which are clustered together) into the same cluster id, we can guarantee k-anonymity among them (more details in Section 4.2). Ultimately, quantization further reduces our problem to *learning distributions over sequences of discrete values*, namely the sequences of cluster ids of feature vectors in each computation graph. Such a problem is naturally addressed by Transformers, state-of-the-art sequence generative models (Vaswani et al., 2017). In Section 4, we introduce the Computational Graph Transformer (CGT), a novel architecture which (at inference time) generates a new sequence of cluster ids, which are then de-quantized as the mean feature vector of the cluster.

## 3.5 END-TO-END FRAMEWORK FOR A BENCHMARK GRAPH GENERATION PROBLEM

Figure 2 summarizes the entire process of mapping a graph generation problem into a discrete sequence generation problem. In the training phase, we 1) sample a set of computation graphs from the input graph, 2) encode each computation graph using the duplicate encoding scheme to fix adjacency matrices, 3) quantize feature vectors to cluster ids they belong to, and finally 4) hand over a set of *(sequence of cluster ids, node label)* pairs to our new Transformer architecture to learn their distribution. In the generation phase, we follow the same process in the opposite direction: 1) the trained Transformer outputs a set of *(sequence of cluster ids, node label)* pairs, 2) we de-quantize cluster ids back into the feature vector space by replacing them with the mean feature vector of the cluster, 3) we regenerate a computation graph from each sequence of feature vectors with the adjacency matrix fixed by the duplicate encoding scheme, and finally 4) we feed the set of generated computation graphs into the GNN model we want to train or evaluate.

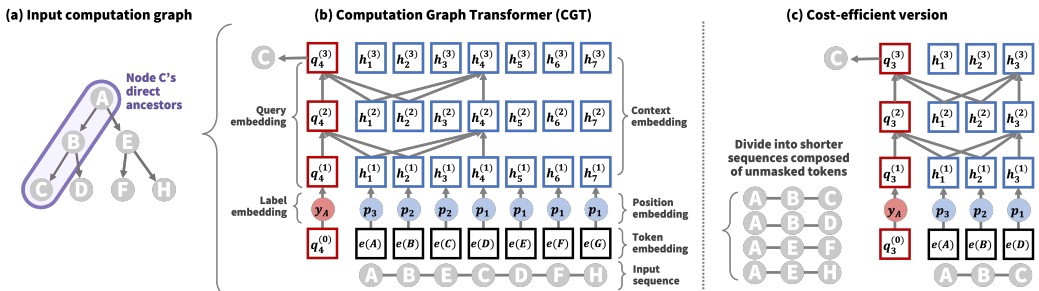

Figure 3: **Computation Graph Transformer (CGT):** (a,b) Given a sequence flattened from the input computation graph, CGT generates context in the forward direction. $e(s_t)$, $q_t^{(l)}$, and $h_t^{(l)}$ denote the token, query, and context embedding of $t$-th token at the $l$-th layer; $p_{l(t)}$ and $y_{s_1}$ denote the position embeddings of $t$-th token and label embedding of the whole sequence, respectively. (c) The cost-efficient version of CGT divides the input sequence into shorter ones composed only of direct ancestor nodes.

# 4 MODEL

We present the Computation Graph Transformer (CGT) that casts computation graph generation as conditional sequence modeling with minimal modification to the Transformer architecture. Then we check our model satisfies the privacy and scalability requirements from Problem Definition 1.

## 4.1 COMPUTATION GRAPH TRANSFORMER

In this work, we extend a two-stream self-attention mechanism, XLNet (Yang et al., 2019), which modifies the Transformer architecture (Vaswani et al., 2017) with a causal self-attention mask to enable auto-regressive generation. Given a sequence $\mathbf{s} = [s_1, \cdots, s_T]$, the $M$-layered Transformer maximizes the likelihood under the forward auto-regressive factorization as follows:

$$\max_{\theta} \log p_\theta(\mathbf{s}) = \sum_{t=1}^{T} \log p_\theta(s_t|\mathbf{s}_{<t}) = \sum_{t=1}^{T} \log \frac{exp(q_\theta^{(L)}(\mathbf{s}_{1:t-1})^\top e(s_t))}{\sum_{s' \neq s_t} exp(q_\theta^{(L)}(\mathbf{s}_{1:t-1})^\top e(s'))}$$

where token embedding $e(s_t)$ maps discrete input id $s_t$ to a randomly initialized trainable vector, and query embedding $q_\theta^{(L)}(\mathbf{s}_{1:t-1})$ encodes information until $(t-1)$-th token in the sequence. More details on the Transformer architecture can be found in the Appendix A.8. Here we describe how we modify XLNet to model our computation graph effectively.

**Position embeddings:** In the original architecture, each token receives a position embedding to let the Transformer recognize the token's position in the sequence. In our model, however, sequences are flattened computation graphs (e.g., input computation graph in Figure 3(a) is flattened into input sequence in Figure 3(b)). To encode the original computation graph structure, we provide different position embeddings to different layers in the computation graph, while nodes at the same layer share the same position embedding. When $l(t)$ denotes the layer number where $t$-th token (node) is located at the original computation graph, position embedding $p_{l(t)}$ indexed by the layer number is assigned to $t$-th token. In Figure 3(b), node $C, D, F$ and $H$ located at the 1-st layer in the computation graph have the same position embedding $p_1$.

**Attention Masks:** In the original architecture, query and context embeddings, $q_t^{(l)}$ and $h_t^{(l)}$, attend to all context embeddings $\mathbf{h}_{1:t-1}^{(l-1)}$ before $t$. In the computation graph, each node is sampled based on its parent node (which is sampled based on its own parent nodes) and is not directly affected by its sibling nodes. To encode this relationship more effectively, we mask all nodes except direct ancestor nodes in the computation graph, i.e., the root node and any nodes between the root node and the leaf node. In Figure 3(b), node $C$'s context/query embeddings attend only to direct ancestors, nodes $A$ and $B$. Note that the number of unmasked tokens are fixed to $L$ in our architecture because there are always $L-1$ direct ancestors in $L$-layered computation graphs. Based on this observation, we provide cost-efficient version of CGT that has shorter sequence length and preserves XLNet's auto-regressive masking as shown in Figure 3(c).

**Label conditioning:** Distributions of neighboring nodes are not only affected by each node's feature information but also by its label. It is well-known that GNNs improve over MLP performance by adding convolutional operations that augment each node's features with neighboring node features. This improvement is commonly attributed to nodes whose feature vectors are noisy (outliers among nodes with the same label) but that are connected with "good" neighbors (whose features are well-aligned with the label). In this case, without label information, we cannot learn whether a node has feature-wise homogeneous neighbors or feature-wise heterogeneous neighbors but with the same labels. In our Transformer model, query embeddings $q_t^{(0)}$ are initialized with label embeddings $y_{s_1}$ that encode the label of the root node $s_1$.

## 4.2 THEORETICAL ANALYSIS

First, our framework can be easily extended to provide $k$-anonymity for node attributes and edge distributions by using k-means clustering with the minimum cluster size $k$ (Bradley et al., 2000) during the quantization phase. Note that we define edge distributions as neighboring node distributions of each node. The full proofs for the following claims can be found in Appendix A.1.

**Claim 1** ($k$-anonymity for node attributes and edge distributions). *In the generated computation graphs, each node attribute and edge distribution appear at least $k$ times, respectively.*

Next, we can provide differential privacy (DP) for node attributes and edge distributions by exploiting DP k-means clustering (Chang et al., 2021) during the quantization phase and DP stochastic gradient descent (DP-SGD) (Song et al., 2013) to train the Transformer. Unfortunately, however, DP-SGD for transformer is not well developed yet practically. So we can not guarantee the *rigid* DP for edge distributions in practice (experimental results in Section 5.4 and more analysis in Appendix A.1). Thus, here, we claim DP only for node attributes.

**Claim 2** (($\epsilon, \delta$)-Differential Privacy for node attributes). *With probability at least $1-\delta$, our generative model $A$ gives $\epsilon$-differential privacy for any graph $\mathcal{G}$, any neighboring graph $\mathcal{G}_{-v}$ without any node $v \in \mathcal{G}$, and any new computation graph $\mathcal{G}_{cg}$ generated from our model as follows:*

$$e^{-\epsilon} \leq \frac{Pr[A(\mathcal{G}) = \mathcal{G}_{cg}]}{Pr[A(\mathcal{G}_{-v}) = \mathcal{G}_{cg}]} \leq e^{\epsilon}$$

Finally, we show that CGT satisfies the scalability requirement in Problem Definition 1:

**Claim 3** (Scalability). *When we aim to generate $L$-layered computation graphs with neighbor sampling number $s$ on a graph with $n$ nodes, computational complexity of CGT training is $O(s^{2L}n)$, and the cost-efficient version is $O(L^2 s^L n)$.*

## 5 EXPERIMENTS

We aim to show that (1) CGT scales to learn distributions of large-scale real-world graphs; (2) CGT generates synthetic graphs on which GNNs perform similarly to the original graphs; (3) synthetic graphs generated by CGT are sufficiently private; and (4) CGT preserves distributions of graph statistics defined on the original set of computation graphs.

## 5.1 EXPERIMENTAL SETTING

We evaluate on seven public datasets — three citation networks (Cora, Citeseer, and Pubmed) (Sen et al., 2008), two co-purchase graphs (Amazon Computer and Amazon Photo) (Shchur et al., 2018), and two co-authorship graph (MS CS and MS Physic) (Shchur et al., 2018). To measure GNN performance similarity, we run popular GNN architectures on a pair of original and synthetic graphs, then measure Pearson and Spearman correlations (Myers et al., 2013) between the resultant performance metrics on each type of graph.

## 5.2 SCALABILITY

To the best of our knowledge, no other generic graph generative model was designed to output a triad of the adjacency matrix, node attribute matrix, and node labels. We extend two VAE-based graph generative models, GVAE (Kipf & Welling, 2016b) and Graphite (Grover et al., 2019) to generate

Table 1: **GNN performance on original and generated graphs in three different scenarios with three variations.** # NE denotes number of noise edges and $\alpha$ denotes the PPR coefficient used for distribution shift.

| | | Aggregation with noise (Citeseer) | | | | Sampling with noise (Amazon Photo) | | | | Distribution shift (MS Physic) | |
|---|---|---|---|---|---|---|---|---|---|---|---|
| **#NE** | **model** | **Original** | **Generated** | **#NE** | **model** | **Original** | **Generated** | $\alpha$ | **model** | **Original** | **Generated** |
| 0 | GCN | 0.73+0.004 | 0.59+0.024 | 0 | GSage | 0.75+0.009 | 0.53+0.028 | iid | GSage | 0.93+0.002 | 0.84+0.008 |
| | SGC | 0.73+0.002 | 0.58+0.029 | | AS-GCN | 0.14+0.016 | 0.12+0.020 | | SGC | 0.92+0.001 | 0.84+0.007 |
| | GIN | 0.71+0.009 | 0.57+0.028 | | FastGCN | 0.92+0.004 | 0.87+0.002 | | GAT | 0.93+0.002 | 0.82+0.011 |
| | GAT | 0.71+0.003 | 0.57+0.029 | | PASS | 0.85+0.011 | 0.54+0.049 | | PPNP | 0.93+0.005 | 0.84+0.007 |
| 2 | GCN | 0.57+0.005 | 0.46+0.013 | 2 | GSage | 0.40+0.012 | 0.36+0.009 | 0.01 | GSage | 0.83+0.033 | 0.76+0.019 |
| | SGC | 0.57+0.005 | 0.47+0.019 | | AS-GCN | 0.12+0.014 | 0.11+0.027 | | SGC | 0.84+0.004 | 0.74+0.015 |
| | GIN | 0.54+0.020 | 0.44+0.015 | | FastGCN | 0.87+0.005 | 0.81+0.010 | | GAT | 0.87+0.007 | 0.78+0.009 |
| | GAT | 0.57+0.014 | 0.44+0.010 | | PASS | 0.73+0.018 | 0.59+0.011 | | PPNP | 0.84+0.007 | 0.74+0.009 |
| 4 | GCN | 0.51+0.027 | 0.41+0.003 | 4 | GSage | 0.26+0.009 | 0.20+0.014 | 0.3 | GSage | 0.84+0.012 | 0.68+0.023 |
| | SGC | 0.52+0.009 | 0.41+0.005 | | AS-GCN | 0.10+0.025 | 0.13+0.054 | | SGC | 0.81+0.009 | 0.70+0.009 |
| | GIN | 0.48+0.023 | 0.41+0.007 | | FastGCN | 0.67+0.003 | 0.62+0.006 | | GAT | 0.85+0.011 | 0.72+0.019 |
| | GAT | 0.49+0.012 | 0.40+0.009 | | PASS | 0.64+0.017 | 0.50+0.017 | | PPNP | 0.81+0.012 | 0.70+0.009 |
| **Pearson / Spearman** | | 0.991 / 0.964 | | | | 0.958 / 0.916 | | | | 0.925 / 0.815 | |

Table 2: **Benchmark effectiveness on node classification.** Pearson and Spearman scores measure the correlation in ranking of GNN models on original and generated graphs; a score of 1 denotes perfect correlation.

| Dataset | Aggregation with noise | | Sampling with noise | | Sampling number | | Distribution shift | |
|---|---|---|---|---|---|---|---|---|
| | Pearson | Spearman | Pearson | Spearman | Pearson | Spearman | Pearson | Spearman |
| Cora | 0.934 | 0.950 | 0.943 | 0.894 | 0.967 | 0.814 | 0.867 | 0.833 |
| Citeseer | 0.991 | 0.964 | 0.955 | 0.977 | 0.973 | 0.904 | 0.812 | 0.799 |
| Pubmed | 0.818 | 0.791 | 0.900 | 0.867 | 0.989 | 0.824 | 0.830 | 0.794 |
| Amazon Computer | 0.825 | 0.778 | 0.885 | 0.916 | 0.975 | 0.890 | 0.906 | 0.860 |
| Amazon Photo | 0.918 | 0.893 | 0.958 | 0.916 | 0.961 | 0.931 | 0.771 | 0.847 |
| MS CS | 0.916 | 0.922 | 0.974 | 0.956 | 0.986 | 0.901 | 0.792 | 0.751 |
| MS Physic | 0.661 | 0.685 | 0.956 | 0.951 | 0.947 | 0.901 | 0.925 | 0.815 |

node attributes and labels in addition to adjacency matrices from their latent variables. We also choose three molecule graph generative models, GraphAF (Shi et al., 2020), GraphDF (Luo et al., 2021), and GraphEBM (Suhail et al., 2021), that do not rely on any molecule-specific traits (e.g., SMILES representation). GraphAF, GraphDF, and Graphite meet out-of-memory errors on even the smallest dataset, Cora (Table 5 in Appendix A.2). This is not surprising, given they were originally designed for small-size molecule graphs. The remaining baselines (GVAE and GraphEBM), however, fail to learn any meaningful node attribute/label distributions from the original graphs. For instance, the predicted distribution sometimes collapses to generating the the same node feature/labels across all nodes, which is obviously not the most effective benchmark (100% accuracy for all GNN models). We show their results and our analysis in the Appendix A.2. Only our method can successfully generate benchmark graphs across all datasets with meaningful node attribute/label distributions (Tables 1 and 2).

## 5.3 BENCHMARK EFFECTIVENESS

To examine the benchmark effectiveness of our generative model, we design 4 different scenarios where the performance of different GNN architectures varies widely. In each scenario, we provide 3 variations to graphs and run 4 GNN models on each variation. For each scenario-variation, we report Pearson and Spearman correlations of the GNN performance metrics on the original graph against those on the generated graph. Due to the space limitation, we present detailed GNN accuracies only on a few datasets/scenarios in Table 1. Results on other datasets can be found in Appendix A.3. Note that Table 2 presents the correlation coefficients across all datasets and scenarios. Descriptions of each GNN model can be found in the Appendix A.7.1.

**SCENARIO 1: noisy edges on aggregation strategies.** We choose four different GNN models with different aggregation strategies: GCN (Kipf & Welling, 2016a) with mean aggregator, GIN (Xu et al., 2018) with sum aggregator, SGC (Wu et al., 2019) with linear aggregator, and GAT (Veličković et al., 2017) with attention aggregator. Then we modify the graph by adding different numbers of noisy edges (randomly connected with any node in the graph) to each node and check how the GNN performance changes. In Table 1, first three columns show the result in the Citeseer dataset. When more noisy edges are added, the accuracy across all GNN models drops in the original graphs. These trends can be nearly exactly captured in GNN performance on the generated graphs (both Pearson and Spearman correlation rates are up to 0.964). This shows that the synthetic graphs generated by our method successfully capture the noisy edge distributions introduced in the original graphs.

**SCENARIO 2: noisy edges on neighbor sampling.** We choose four different GNN models with different neighbor sampling strategies: GraphSage (abbreviated as GSage in Table 1) (Hamilton et al., 2017) with random sampling, FastGCN (Chen et al., 2018) with heuristic layer-wise sampling, AS-GCN (Huang et al., 2018) with trainable layer-wise sampling, and PASS (Yoon et al., 2021) with trainable node-wise sampling. We then add noisy edges as described in the ENV 1 and check how the different sampling policies deal with noisy neighbors. In Table 1, FastGCN shows highest accuracies across different number of nosiy edges, followed by PASS, GraphSage, and AS-GCN on the original graphs on the Amazon Photo dataset; and this trend is well-preserved on the generated graphs, showing 0.958 Pearson correlation.

**SCENARIO 3: different sampling numbers on neighbor sampling.** We choose the same four GNN models with different neighbor sampling strategies as in EV2. Then we change the number of sampled neighbor nodes and check how the GNN performance is affected. As shown in Table 2, trends among original graphs — more neighbors are sampled, GNN performance generally increases — are successfully captured in the generated graphs with 0.991 and 0.931 Pearson and Spearman correlations, respectively, across all datasets. You can find the detailed GNN accuracies in Appendix A.3.

Table 3: **Benchmark effectiveness and GNN performance on link prediction.**

| Dataset | Pearson | Spearman | Pearson | Spearman |
|---|---|---|---|---|
| Cora | 0.781 | 0.741 | | |
| Citeseer | 0.808 | 0.824 | | |
| Pubmed | 0.725 | 0.420 | 0.742 | 0.754 |
| AmazonC | 0.652 | 0.559 | | |
| AmazonP | 0.887 | 0.443 | | |

| Dataset | Predictor | Model | Original | Generated |
|---|---|---|---|---|
| Citeseer | Dot | GCN | 0.69+0.007 | 0.65+0.026 |
| | | SGC | 0.70+0.003 | 0.67+0.022 |
| | | GIN | 0.83+0.008 | 0.65+0.010 |
| | | GAT | 0.75+0.005 | 0.68+0.021 |
| | MLP | GCN | 0.58+0.005 | 0.59+0.010 |
| | | SGC | 0.58+0.008 | 0.59+0.023 |
| | | GIN | 0.57+0.011 | 0.61+0.024 |
| | | GAT | 0.61+0.005 | 0.62+0.009 |

**SCENARIO 4: distribution shift.** (Zhu et al., 2021) proposed a biased training set sampler to examine each GNN model's robustness to distribution shift between the training/test time. The biased sampler picks a few seed nodes and finds nearby nodes using the Personalized PageRank vectors (Page et al., 1999) $\pi_{\mathrm{ppr}} = (I - (1-\alpha)\tilde{A})^{-1}$ with decaying coefficient $\alpha$, then uses them to compose a biased training set. We choose the same GNN models, GCN (Kipf & Welling, 2016a), SGC (Wu et al., 2019), GAT (Veličković et al., 2017), and PPNP (Klicpera et al., 2018), as the original paper (Zhu et al., 2021) chose for their baselines. We vary $\alpha$ and check how each GNN models deal with the biased training set. In the last three columns in Table 1, the performance of GNN models drops as $\alpha$ increases, and the generated graphs successfully capture these trends.

**Link prediction.** As nodes are the minimum unit in graphs that compose edges or subgraphs, we can generate subgraphs for edges by merging computation graphs of their component nodes. Here we show link prediction results on original graphs are also preserved successfully on our generated graphs. We run GCN, SGC, GIN, and GAT on graphs, followed by Dot product or MLP to predict link probabilities. Table 3 shows Pearson and Spearman correlations across 8 different combinations of link prediction models (4 GNN models $\times$ 2 predictors) on each dataset and across the whole datasets. The lower table shows the detailed link prediction accuracies on the Citeseer dataset. Our model generates graphs that substitute original graphs successfully, preserving the ranking of GNN link prediction performance with 0.754 Spearman correlation across the datasets.

Table 4: **Privacy-Performance trade-off in graph generation on the Cora dataset**

| | Original | No privacy | K-anonymity | | | DP kmean ($\delta = 0.01$) | | | DP SGD ($\delta = 0.1$) | |
|---|---|---|---|---|---|---|---|---|---|---|
| | | | $k = 100$ | $k = 500$ | $k = 1000$ | $\epsilon = 1$ | $\epsilon = 10$ | $\epsilon = 25$ | $\epsilon = 10^6$ | $\epsilon = 10^9$ |
| Pearson | 1.000 | 0.934 | 0.916 | 0.862 | 0.030 | 0.874 | 0.844 | 0.804 | 0.112 | 0.890 |
| Spearman | 1.000 | 0.935 | 0.947 | 0.812 | 0.018 | 0.869 | 0.805 | 0.807 | 0.116 | 0.959 |

## 5.4 PRIVACY

We examine the performance-privacy trade-off across different privacy guarantees. For $k$-anonymity, we use the k-means clustering algorithm (Bradley et al., 2000) varying the minimum cluster size $k$. For Differential Privacy (DP) for node attributes, we use DP k-means (Chang et al., 2021) varying the privacy cost $\epsilon$ while setting $\delta = 0.01$. As expected, in Table 4, higher $k$ and smaller $\epsilon$ (i.e., stronger privacy) hinder the generative model's ability to learn the exact distributions of the original graphs,

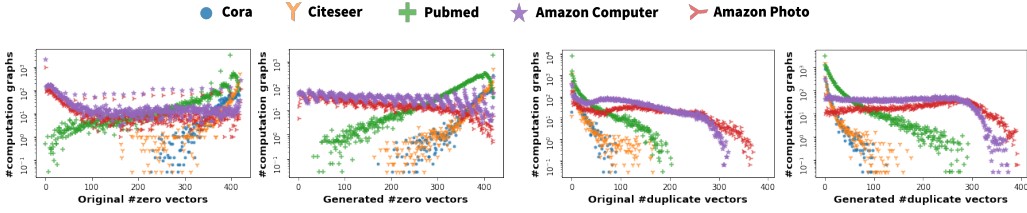

(a) # zero vectors on each computation graph          (b) # redundant vectors on each computation graph

Figure 4: **CGT preserves distributions of graph statistics in generated graphs for each dataset:** Duplicate encoding infuses graph structure into feature matrices of computation graphs. In each computation graph, # zero vectors is inversely proportional to edge density, while # redundant vectors is proportional to # cycles.

and the GNN performance gaps between original and generated graphs increase (lower Pearson and Spearman coefficients). Detailed GNN accuracies could be found in Table 12 in Appendix A.4. To provide DP for edge distributions, we use DP stochastic gradient descent (Song et al., 2013) to train the transformer, varying the privacy cost $\epsilon$ while setting $\delta = 0.1$. In Table 4, even with astronomically low privacy cost ($\epsilon = 10^6$), the performance of our generative model degrades significantly. When we set $\epsilon = 10^9$ (which is impractical), we can finally see a reasonable performance. This shows the limited performance of DP SGD on the transformer architecture.

## 5.5 Graph Statistics

Given a source graph, our method generates a set of computation graphs without any node ids. In other words, attackers cannot merge the generated computation graphs to restore the original graph and re-identify node information. Thus, instead of traditional graph statistics such as orbit counts or clustering coefficients that rely on the global view of graphs, we define new graph statistics for computation graphs that are encoded by the duplicate scheme. Duplicate scheme fixes adjacency matrices across all computation graphs by infusing structural information (originally encoded in adjacency matrices) into feature matrices. *Number of zero vectors* (corresponding to null nodes that are padded when a node has fewer neighbors than a sampling neighbor number) is inversely proportional to edge density in a computation graph. *Number of duplicate feature vectors* (which are duplicated when nodes share neighbors) is proportional to number (#) of cycles in a computation graph, indicating how densely nodes are connected in the computation graph. In Figure 4, we present distributions of a) number of zero vectors and b) number of redundant vectors each computation graph has on each dataset. Five different datasets show distinct distributions on these two statistics, and distributions of both statistics are well preserved in generated graphs. For instance, in Figure 4(a), green lines (Pubmed) are lower than purple and red lines (Amazon Computer and Amazon Photo) at the beginning and become higher in both plots. In Figure 4(b), purple lines (Amazon Computer) are slightly higher than red lines (Amazon Photo) until $x = 300$, then become lower in both plots. In the same figure, blue, yellow, and green lines (Cora, Citeseer, and Pubmed) decrease sharply compared to purple and red lines (Amazon Computer and Amazon Photo) in both plots. This shows our generative model preserves graph structures encoded in feature matrices successfully.

## 6 Conclusion

We propose a new graph generation problem to enable generating benchmark graphs for GNNs that follow distributions of (possibly proprietary) source graphs with three requirements: 1) benchmark effectiveness, 2) privacy guarantee, and 3) scalability. With a novel graph encoding scheme, we reframe a large-scale graph generation problem into a medium-length sequence generation problem and apply the strong generation power of the Transformer architecture to the graph domain.

**Limitation of the study:** This paper shows that clustering-based solutions can achieve k-anonymity privacy guarantees. We stress, however, that implementing a real-world system with strong privacy guarantees will need to consider many other aspects beyond the scope of this paper. We leave as future work the study of whether we can combine stronger privacy guarantees with those of k-anonymity to enhance privacy protection.

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

# A    APPENDIX

## A.1    PROOF OF PRIVACY CLAIMS

**Claim 1** ($k$-Anonymity for node attributes and edge distributions). *In the generated computation graphs, each node attribute and edge distribution appear at least $k$ times, respectively.*

*Proof.* In the quantization phase, we use the k-means clustering algorithm (Bradley et al., 2000) with a minimum cluster size $k$. Then each node id is replaced with the id of the cluster it belongs to, reducing the original $(n \times n)$ graph into a $(m \times m)$ hypergraph where $m = n/k$ is the number of clusters. Then Computation Graph Transformer learns edge distributions among $m$ hyper nodes (i.e., clusters) and generates a new $(m \times m)$ hypergraph. In the hypergraph, there are at most $m$ different node attributes and $m$ different edge distributions. During the de-quantization phase, a $(m \times m)$ hypergraph is mapped back to a $(n \times n)$ graph by letting $k$ nodes in each cluster follow their cluster's node attributes/edge distributions as follows: $k$ nodes in the same cluster will have the same feature vector that is the average feature vector of original nodes belonging to the cluster. When $s$ denotes the number of sampled neighbor nodes, each node samples $s$ clusters (with replacement) following its cluster's edge distributions among $m$ clusters. When a node samples cluster $i$, it will be connected to one of nodes in the cluster $i$ randomly. At the end, each node will have $s$ neighbor nodes randomly sampled from $s$ clusters the node samples with the cluster's edge distribution, respectively. Likewise, all $k$ nodes belonging to the same cluster will sample neighbors following the same edge distributions. Thus each node attribute and edge distribution appear at least $k$ times in a generated graph. ∎

**Claim 2** (($\epsilon, \delta$)-Differential Privacy for node attributes). *With probability at least $1 - \delta$, our generative model $A$ gives $\epsilon$-differential privacy for any graph $\mathcal{G}$, any neighboring graph $\mathcal{G}_{-v}$ without any node $v \in \mathcal{G}$, and any new computation graph $\mathcal{G}_{cg}$ generated from our model as follows:*

$$e^{-\epsilon} \leq \frac{Pr[A(\mathcal{G}) = \mathcal{G}_{cg}]}{Pr[A(\mathcal{G}_{-v}) = \mathcal{G}_{cg}]} \leq e^{\epsilon}$$

*Proof.* $\mathcal{G}_{-v}$ denotes neighboring graphs to the original one $\mathcal{G}$, but without a specific node $v$. During the quantization phase, we use $(\epsilon, \delta)$-differential private k-means clustering algorithm on node features (Chang et al., 2021). Then clustering results are differentially private with regard to each node features. In the generated graphs, each node feature is decided by the clustering results (i.e., the average feature vector of nodes belonging to the same cluster). Then, by looking at the generated node features, one cannot tell whether any individual node feature was included in the original dataset or not. ∎

**Remark 1**    (($\epsilon, \delta$)-Differential Privacy for edge distributions). In our model, individual nodes' edge distributions are learned and generated by the transformer. When we use $(\epsilon, \delta)$-differential private stochastic gradient descent (DP-SGD) (Song et al., 2013) to train the transformer, the transformer becomes differentially private in the sense that by looking at the output (generated edge distributions), one cannot tell whether any individual node's edge distribution (input to the transformer) was included in the original dataset or not. If we have DP-SGD that can train transformers successfully with reasonably small $\epsilon$ and $\delta$, we can guarantee $(\epsilon, \delta)$-differential privacy for edge distribution of any graph generated by our generative model. However, as we show in Section 5.4, current DP-SGD is not stable yet for transformer training, leading to very coarse or impractical privacy guarantees.

**Claim 3** (Scalability). *When we aim to generate $L$-layered computation graphs with neighbor sampling number $s$ on a graph with $n$ nodes, computational complexity of* CGT *training is $O(s^{2L}n)$, and that of the cost-efficient version is $O(L^2 s^L n)$.*

*Proof.* During k-means, we randomly sample $n_k$ node features to compute the cluster centers. Then we map each feature vector to the closest cluster center. By sampling $n_k$ nodes, we limit the k-mean computation cost to $O(n_k^2)$. The sequence flattened from each computation graph is $O(1 + s + \cdots + s^L)$ and the number of sequences (computation graphs) is $O(n)$. Then the training time of the transformer is proportional to $O(s^{2L}n)$. In total, the complexity is $O(s^{2L}n + n_k^2)$. As $s^{2L}n >> n_k^2$, the final computation complexity becomes $O(s^{2L}n)$. In the cost-efficient version, the length of sequences (composed only of direct ancestor nodes) is reduced to $L$. However, the

Table 5: **GNN performance on graphs generated by baseline generative models.** Except our method, no existing graph generative models can generate a set of adjacency matrix, node feature matrix, and node label matrix that reproduce reasonable GNN performance.

| Dataset | Model | Original | GraphAF | GraphDF | GraphEBM | Graphite | GVAE | Ours |
|---------|-------|----------|---------|---------|----------|----------|------|------|
| Cora | GCN | 0.860 | o.o.m | o.o.m | 1.000 | o.o.m | 0.200 | 0.760 |
| | SGC | 0.850 | o.o.m | o.o.m | 1.000 | o.o.m | 0.200 | 0.750 |
| | GIN | 0.850 | o.o.m | o.o.m | 1.000 | o.o.m | 0.200 | 0.750 |
| | GAT | 0.830 | o.o.m | o.o.m | 1.000 | o.o.m | 0.380 | 0.750 |
| Citeseer | GCN | 0.730 | o.o.m | o.o.m | 1.000 | o.o.m | 0.190 | 0.590 |
| | SGC | 0.730 | o.o.m | o.o.m | 1.000 | o.o.m | 0.180 | 0.580 |
| | GIN | 0.710 | o.o.m | o.o.m | 1.000 | o.o.m | 0.190 | 0.570 |
| | GAT | 0.710 | o.o.m | o.o.m | 1.000 | o.o.m | 0.380 | 0.570 |
| Pearson | | 1 | - | - | - | - | -0.062 | 0.993 |
| Spearman | | 1 | - | - | - | - | 0.195 | 0.975 |

number of sequences increases to $s^L n$ because each nodes has one computation graph composed of $s^L$ shortened sequences. Then the final computation complexity become $O(L^2 s^L n)$. ∎

## A.2 SCALABILITY ISSUES ON BASELINE GENERATIVE MODELS

In Table 5, GraphAF (Shi et al., 2020), GraphDF (Luo et al., 2021), and Graphite (Grover et al., 2019) meet out-of-memory errors on both the Cora and Citeseer datasets. This is because they were originally designed for small-size molecule graphs. The remaining baselines, GVAE (Kipf & Welling, 2016b) and GraphEBM (Suhail et al., 2021) successfully generate graphs, however, fail to learn any meaningful node attribute/label distributions from the original graphs. Especially, GraphEBM generates graphs whose distribution collapses to generating the the same node feature/labels across all nodes, showing $100\%$ accuracy for all GNN models, which is obviously not the most effective benchmark. Note that none of existing graph generative models is designed for GNN benchmarking — simultaneous generation of adjacency, node feature, and node label matrices, rather they all focus only on the generation of adjacency matrices. This result shows the tricky aspects of graph generation and relations among graph structure, node attributes and labels, and a large room for improvement in the graph generation field.

## A.3 DETAILED GNN PERFORMANCE IN THE BENCHMARK EFFECTIVE EXPERIMENT IN SECTION 5.3

Tables 6, 7, 8, and 9 show GNN performance on node classification tasks across the original/quantized/generated graphs. Quantized graphs are graphs after the quantization process: each feature vector is replaced by the mean feature vector of a cluster it belongs to, and adjacency matrices are a constant encoded by the duplicate encoding scheme. Quantized graphs are input to CGT, and generated graphs are output from CGT as presented in Figure 2. Likewise, Table 10 shows GNN performance on link prediction tasks across the original/quantized/generated graphs. As presented across all five Tables, our proposed generative model CGT successfully generates synthetic substitutes of large-scale real-world graphs that shows similar task performance as on the original graphs. Table 11 shows benchmark effective across all 9 GNN models we have used in the experiments on the original graphs without any variations.

## A.4 DETAILED GNN PERFORMANCE IN THE PRIVACY EXPERIMENT IN SECTION 5.4

Table 12 shows detailed privacy-GNN performance trade-off on the Cora dataset. In K-anonymity, higher k (i.e., more nodes in the same clusters, thus stronger privacy) hinders the generative model's ability to learn the exact distributions of the original graphs, and the GNN performance gaps between original and generated graphs increase, showing lower Pearson and Spearman coefficients. In DP kmeans, smaller $\epsilon$ (i.e., higher privacy cost) results in lower GNN performance, however, surprisingly, showing higher Pearson and Spearman coefficients. This is because DP kmeans could remove noises in graphs (while hiding outliers for privacy) and capture representative distributions on the original graph more easily. This results show our Claims 1 and 2 on privacy are holding on real-world experiments. As we discussed in Section 4.2, DP SGD fails to train the transformer, showing low GNN performance even with astronomically low privacy cost ($\epsilon = 10^6$).

Table 6: **GNN performance on SCENARIO 1: noisy edges on aggregation strategies.**

| Dataset | #NE | model | Original | std | Cluster | std | Generated | std | pearson | spearman |
|---|---|---|---|---|---|---|---|---|---|---|
| Cora | 0 | GCN | 0.860 | 0.002 | 0.830 | 0.002 | 0.760 | 0.005 | | |
| | | SGC | 0.850 | 0.001 | 0.820 | 0.004 | 0.750 | 0.002 | | |
| | | GIN | 0.850 | 0.004 | 0.830 | 0.008 | 0.750 | 0.013 | | |
| | | GAT | 0.830 | 0.002 | 0.830 | 0.002 | 0.750 | 0.006 | | |
| | 2 | GCN | 0.770 | 0.008 | 0.750 | 0.009 | 0.680 | 0.014 | 0.934 | 0.950 |
| | | SGC | 0.770 | 0.008 | 0.740 | 0.003 | 0.680 | 0.015 | | |
| | | GIN | 0.780 | 0.002 | 0.730 | 0.003 | 0.670 | 0.009 | | |
| | | GAT | 0.680 | 0.013 | 0.740 | 0.002 | 0.660 | 0.009 | | |
| | 4 | GCN | 0.720 | 0.011 | 0.690 | 0.008 | 0.610 | 0.015 | | |
| | | SGC | 0.720 | 0.005 | 0.690 | 0.004 | 0.600 | 0.007 | | |
| | | GIN | 0.660 | 0.019 | 0.680 | 0.007 | 0.590 | 0.016 | | |
| | | GAT | 0.600 | 0.019 | 0.670 | 0.008 | 0.570 | 0.015 | | |
| **Dataset** | **#NE** | **model** | **Original** | **std** | **Cluster** | **std** | **Generated** | **std** | **pearson** | **spearman** |
| Citeseer | 0 | GCN | 0.730 | 0.004 | 0.680 | 0.002 | 0.590 | 0.024 | | |
| | | SGC | 0.730 | 0.002 | 0.670 | 0.002 | 0.580 | 0.029 | | |
| | | GIN | 0.710 | 0.009 | 0.670 | 0.004 | 0.570 | 0.028 | | |
| | | GAT | 0.710 | 0.003 | 0.670 | 0.004 | 0.570 | 0.029 | | |
| | 2 | GCN | 0.570 | 0.005 | 0.560 | 0.010 | 0.460 | 0.013 | 0.991 | 0.964 |
| | | SGC | 0.570 | 0.005 | 0.570 | 0.007 | 0.470 | 0.019 | | |
| | | GIN | 0.540 | 0.020 | 0.560 | 0.003 | 0.440 | 0.015 | | |
| | | GAT | 0.570 | 0.014 | 0.550 | 0.004 | 0.440 | 0.01 | | |
| | 4 | GCN | 0.510 | 0.027 | 0.500 | 0.001 | 0.410 | 0.003 | | |
| | | SGC | 0.520 | 0.009 | 0.500 | 0.002 | 0.410 | 0.005 | | |
| | | GIN | 0.480 | 0.023 | 0.510 | 0.008 | 0.410 | 0.007 | | |
| | | GAT | 0.490 | 0.012 | 0.510 | 0.004 | 0.400 | 0.009 | | |
| **Dataset** | **#NE** | **model** | **Original** | **std** | **Cluster** | **std** | **Generated** | **std** | **pearson** | **spearman** |
| Pubmed | 0 | GCN | 0.860 | 0.001 | 0.820 | 0.001 | 0.780 | 0.007 | | |
| | | SGC | 0.860 | 0.000 | 0.810 | 0.001 | 0.780 | 0.003 | | |
| | | GIN | 0.830 | 0.006 | 0.810 | 0.001 | 0.770 | 0.002 | | |
| | | GAT | 0.860 | 0.002 | 0.820 | 0.003 | 0.780 | 0.005 | | |
| | 2 | GCN | 0.780 | 0.004 | 0.760 | 0.004 | 0.680 | 0.003 | 0.818 | 0.791 |
| | | SGC | 0.760 | 0.004 | 0.750 | 0.006 | 0.670 | 0.004 | | |
| | | GIN | 0.790 | 0.012 | 0.740 | 0.014 | 0.670 | 0.007 | | |
| | | GAT | 0.710 | 0.011 | 0.770 | 0.003 | 0.680 | 0.005 | | |
| | 4 | GCN | 0.730 | 0.003 | 0.710 | 0.003 | 0.640 | 0.007 | | |
| | | SGC | 0.670 | 0.003 | 0.700 | 0.003 | 0.630 | 0.009 | | |
| | | GIN | 0.770 | 0.011 | 0.700 | 0.008 | 0.600 | 0.017 | | |
| | | GAT | 0.650 | 0.005 | 0.740 | 0.001 | 0.640 | 0.004 | | |
| **Dataset** | **#NE** | **model** | **Original** | **std** | **Cluster** | **std** | **Generated** | **std** | **pearson** | **spearman** |
| Amazon Computer | 0 | GCN | 0.860 | 0.002 | 0.840 | 0.009 | 0.840 | 0.001 | | |
| | | SGC | 0.860 | 0.005 | 0.810 | 0.009 | 0.830 | 0.007 | | |
| | | GIN | 0.850 | 0.002 | 0.810 | 0.015 | 0.800 | 0.013 | | |
| | | GAT | 0.840 | 0.008 | 0.840 | 0.011 | 0.830 | 0.01 | | |
| | 2 | GCN | 0.780 | 0.004 | 0.760 | 0.004 | 0.680 | 0.003 | 0.825 | 0.778 |
| | | SGC | 0.760 | 0.004 | 0.750 | 0.006 | 0.670 | 0.004 | | |
| | | GIN | 0.790 | 0.012 | 0.740 | 0.014 | 0.670 | 0.007 | | |
| | | GAT | 0.710 | 0.011 | 0.770 | 0.003 | 0.680 | 0.005 | | |
| | 4 | GCN | 0.730 | 0.003 | 0.710 | 0.003 | 0.640 | 0.007 | | |
| | | SGC | 0.670 | 0.003 | 0.700 | 0.003 | 0.630 | 0.009 | | |
| | | GIN | 0.770 | 0.011 | 0.700 | 0.008 | 0.600 | 0.017 | | |
| | | GAT | 0.650 | 0.005 | 0.740 | 0.001 | 0.640 | 0.004 | | |
| **Dataset** | **#NE** | **model** | **Original** | **std** | **Cluster** | **std** | **Generated** | **std** | **pearson** | **spearman** |
| Amazon Photo | 0 | GCN | 0.910 | 0.001 | 0.890 | 0.003 | 0.900 | 0.005 | | |
| | | SGC | 0.910 | 0.000 | 0.890 | 0.005 | 0.900 | 0.006 | | |
| | | GIN | 0.900 | 0.003 | 0.880 | 0.005 | 0.900 | 0.001 | | |
| | | GAT | 0.900 | 0.009 | 0.880 | 0.010 | 0.890 | 0.007 | | |
| | 2 | GCN | 0.870 | 0.007 | 0.870 | 0.003 | 0.790 | 0.007 | 0.918 | 0.893 |
| | | SGC | 0.870 | 0.005 | 0.870 | 0.008 | 0.790 | 0.005 | | |
| | | GIN | 0.870 | 0.006 | 0.870 | 0.004 | 0.770 | 0.012 | | |
| | | GAT | 0.860 | 0.006 | 0.860 | 0.005 | 0.780 | 0.003 | | |
| | 4 | GCN | 0.820 | 0.019 | 0.810 | 0.003 | 0.740 | 0.002 | | |
| | | SGC | 0.830 | 0.001 | 0.810 | 0.022 | 0.730 | 0.012 | | |
| | | GIN | 0.840 | 0.006 | 0.830 | 0.009 | 0.710 | 0.024 | | |
| | | GAT | 0.860 | 0.010 | 0.820 | 0.029 | 0.720 | 0.01 | | |
| **Dataset** | **#NE** | **model** | **Original** | **std** | **Cluster** | **std** | **Generated** | **std** | **pearson** | **spearman** |
| MS CS | 0 | GCN | 0.880 | 0.004 | 0.890 | 0.003 | 0.830 | 0.008 | | |
| | | SGC | 0.880 | 0.003 | 0.880 | 0.002 | 0.830 | 0.008 | | |
| | | GIN | 0.870 | 0.001 | 0.870 | 0.004 | 0.820 | 0.013 | | |
| | | GAT | 0.880 | 0.003 | 0.890 | 0.004 | 0.830 | 0.006 | | |
| | 2 | GCN | 0.860 | 0.005 | 0.870 | 0.005 | 0.760 | 0.005 | 0.916 | 0.922 |
| | | SGC | 0.860 | 0.006 | 0.860 | 0.004 | 0.750 | 0.006 | | |
| | | GIN | 0.850 | 0.010 | 0.840 | 0.005 | 0.720 | 0.002 | | |
| | | GAT | 0.860 | 0.007 | 0.860 | 0.005 | 0.750 | 0.01 | | |
| | 4 | GCN | 0.840 | 0.003 | 0.840 | 0.004 | 0.710 | 0.009 | | |
| | | SGC | 0.840 | 0.002 | 0.840 | 0.002 | 0.700 | 0.005 | | |
| | | GIN | 0.820 | 0.009 | 0.790 | 0.010 | 0.670 | 0.011 | | |
| | | GAT | 0.860 | 0.011 | 0.850 | 0.004 | 0.700 | 0.005 | | |
| **Dataset** | **#NE** | **model** | **Original** | **std** | **Cluster** | **std** | **Generated** | **std** | **pearson** | **spearman** |
| MS Physic | 0 | GCN | 0.930 | 0.002 | 0.930 | 0.002 | 0.840 | 0.008 | | |
| | | SGC | 0.920 | 0.001 | 0.920 | 0.001 | 0.840 | 0.007 | | |
| | | GIN | 0.930 | 0.002 | 0.920 | 0.002 | 0.820 | 0.011 | | |
| | | GAT | 0.930 | 0.005 | 0.930 | 0.000 | 0.840 | 0.007 | | |
| | 2 | GCN | 0.910 | 0.000 | 0.910 | 0.001 | 0.770 | 0.004 | 0.661 | 0.685 |
| | | SGC | 0.890 | 0.002 | 0.900 | 0.000 | 0.760 | 0.004 | | |
| | | GIN | 0.910 | 0.009 | 0.900 | 0.002 | 0.750 | 0.008 | | |
| | | GAT | 0.930 | 0.003 | 0.900 | 0.003 | 0.770 | 0.003 | | |
| | 4 | GCN | 0.880 | 0.006 | 0.890 | 0.002 | 0.710 | 0.006 | | |
| | | SGC | 0.860 | 0.003 | 0.880 | 0.002 | 0.710 | 0.005 | | |
| | | GIN | 0.900 | 0.006 | 0.880 | 0.005 | 0.700 | 0.007 | | |
| | | GAT | 0.930 | 0.002 | 0.890 | 0.001 | 0.720 | 0.004 | | |

Table 7: **GNN performance on SCENARIO 2: noisy edges on neighbor sampling.**

| Dataset | #NE | model | Original | std | Cluster | std | Generated | std | pearson | spearman |
|---|---|---|---|---|---|---|---|---|---|---|
| **Cora** | **0** | GraphSage | 0.740 | 0.012 | 0.560 | 0.008 | 0.490 | 0.011 | 0.943 | 0.894 |
| | | AS-GCN | 0.130 | 0.014 | 0.110 | 0.013 | 0.130 | 0.013 | | |
| | | FastGCN | 0.440 | 0.006 | 0.390 | 0.005 | 0.370 | 0.006 | | |
| | | PASS | 0.790 | 0.011 | 0.620 | 0.008 | 0.560 | 0.029 | | |
| | **2** | GraphSage | 0.360 | 0.012 | 0.300 | 0.014 | 0.270 | 0.004 | | |
| | | AS-GCN | 0.130 | 0.013 | 0.110 | 0.010 | 0.130 | 0.016 | | |
| | | FastGCN | 0.320 | 0.010 | 0.290 | 0.007 | 0.280 | 0.008 | | |
| | | PASS | 0.630 | 0.021 | 0.520 | 0.023 | 0.440 | 0.034 | | |
| | **4** | GraphSage | 0.130 | 0.005 | 0.150 | 0.007 | 0.170 | 0.015 | | |
| | | AS-GCN | 0.180 | 0.057 | 0.130 | 0.008 | 0.130 | 0.008 | | |
| | | FastGCN | 0.540 | 0.013 | 0.610 | 0.020 | 0.570 | 0.01 | | |
| | | PASS | 0.560 | 0.008 | 0.520 | 0.003 | 0.400 | 0.016 | | |
| Dataset | #NE | model | Original | std | Cluster | std | Generated | std | pearson | spearman |
| **Citeseer** | **0** | GraphSage | 0.660 | 0.005 | 0.510 | 0.014 | 0.430 | 0.007 | 0.955 | 0.977 |
| | | AS-GCN | 0.100 | 0.006 | 0.100 | 0.019 | 0.090 | 0.006 | | |
| | | FastGCN | 0.380 | 0.011 | 0.330 | 0.009 | 0.300 | 0.001 | | |
| | | PASS | 0.680 | 0.008 | 0.530 | 0.012 | 0.440 | 0.006 | | |
| | **2** | GraphSage | 0.250 | 0.003 | 0.310 | 0.005 | 0.280 | 0.005 | | |
| | | AS-GCN | 0.090 | 0.006 | 0.080 | 0.006 | 0.090 | 0.01 | | |
| | | FastGCN | 0.240 | 0.007 | 0.260 | 0.008 | 0.230 | 0.003 | | |
| | | PASS | 0.540 | 0.008 | 0.460 | 0.010 | 0.410 | 0.014 | | |
| | **4** | GraphSage | 0.190 | 0.008 | 0.240 | 0.005 | 0.250 | 0.012 | | |
| | | AS-GCN | 0.110 | 0.012 | 0.100 | 0.021 | 0.100 | 0.004 | | |
| | | FastGCN | 0.210 | 0.004 | 0.210 | 0.006 | 0.200 | 0.014 | | |
| | | PASS | 0.480 | 0.021 | 0.460 | 0.002 | 0.400 | 0.015 | | |
| Dataset | #NE | model | Original | std | Cluster | std | Generated | std | pearson | spearman |
| **Pubmed** | **0** | GraphSage | 0.780 | 0.005 | 0.680 | 0.002 | 0.630 | 0.004 | 0.885 | 0.916 |
| | | AS-GCN | 0.260 | 0.009 | 0.230 | 0.026 | 0.240 | 0.007 | | |
| | | FastGCN | 0.470 | 0.003 | 0.450 | 0.003 | 0.430 | 0.003 | | |
| | | PASS | 0.850 | 0.007 | 0.730 | 0.001 | 0.680 | 0.007 | | |
| | **2** | GraphSage | 0.409 | 0.002 | 0.467 | 0.012 | 0.431 | 0.004 | | |
| | | AS-GCN | 0.308 | 0.072 | 0.419 | 0.053 | 0.287 | 0.051 | | |
| | | FastGCN | 0.731 | 0.008 | 0.727 | 0.008 | 0.628 | 0.008 | | |
| | | PASS | 0.812 | 0.007 | 0.697 | 0.000 | 0.587 | 0.008 | | |
| | **4** | GraphSage | 0.310 | 0.001 | 0.320 | 0.003 | 0.320 | 0.003 | | |
| | | AS-GCN | 0.310 | 0.031 | 0.330 | 0.035 | 0.360 | 0.021 | | |
| | | FastGCN | 0.660 | 0.002 | 0.650 | 0.002 | 0.550 | 0.012 | | |
| | | PASS | 0.790 | 0.001 | 0.690 | 0.006 | 0.430 | 0.005 | | |
| Dataset | #NE | model | Original | std | Cluster | std | Generated | std | pearson | spearman |
| **Amazon Computer** | **0** | GraphSage | 0.630 | 0.027 | 0.520 | 0.022 | 0.460 | 0.012 | 0.958 | 0.916 |
| | | AS-GCN | 0.130 | 0.065 | 0.130 | 0.081 | 0.060 | 0.028 | | |
| | | FastGCN | 0.860 | 0.005 | 0.820 | 0.006 | 0.810 | 0.005 | | |
| | | PASS | 0.720 | 0.014 | 0.590 | 0.004 | 0.540 | 0.009 | | |
| | **2** | GraphSage | 0.260 | 0.001 | 0.200 | 0.012 | 0.140 | 0.002 | | |
| | | AS-GCN | 0.190 | 0.063 | 0.040 | 0.002 | 0.050 | 0.012 | | |
| | | FastGCN | 0.750 | 0.005 | 0.710 | 0.001 | 0.640 | 0.004 | | |
| | | PASS | 0.620 | 0.011 | 0.530 | 0.006 | 0.220 | 0.033 | | |
| | **4** | GraphSage | 0.120 | 0.004 | 0.100 | 0.007 | 0.070 | 0.004 | | |
| | | AS-GCN | 0.090 | 0.045 | 0.050 | 0.018 | 0.100 | 0.037 | | |
| | | FastGCN | 0.650 | 0.004 | 0.620 | 0.001 | 0.570 | 0.006 | | |
| | | PASS | 0.540 | 0.024 | 0.470 | 0.014 | 0.120 | 0.019 | | |
| Dataset | #NE | model | Original | std | Cluster | std | Generated | std | pearson | spearman |
| **Amazon Photo** | **0** | GraphSage | 0.750 | 0.009 | 0.670 | 0.017 | 0.530 | 0.028 | 0.958 | 0.916 |
| | | AS-GCN | 0.140 | 0.016 | 0.080 | 0.025 | 0.120 | 0.02 | | |
| | | FastGCN | 0.920 | 0.004 | 0.900 | 0.003 | 0.870 | 0.002 | | |
| | | PASS | 0.850 | 0.011 | 0.780 | 0.006 | 0.540 | 0.049 | | |
| | **2** | GraphSage | 0.400 | 0.012 | 0.370 | 0.007 | 0.360 | 0.009 | | |
| | | AS-GCN | 0.120 | 0.014 | 0.140 | 0.041 | 0.110 | 0.027 | | |
| | | FastGCN | 0.870 | 0.005 | 0.880 | 0.003 | 0.810 | 0.01 | | |
| | | PASS | 0.730 | 0.018 | 0.640 | 0.029 | 0.590 | 0.011 | | |
| | **4** | GraphSage | 0.260 | 0.009 | 0.200 | 0.016 | 0.200 | 0.014 | | |
| | | AS-GCN | 0.100 | 0.025 | 0.130 | 0.037 | 0.130 | 0.054 | | |
| | | FastGCN | 0.670 | 0.003 | 0.670 | 0.006 | 0.620 | 0.006 | | |
| | | PASS | 0.640 | 0.017 | 0.600 | 0.005 | 0.500 | 0.017 | | |
| Dataset | #NE | model | Original | std | Cluster | std | Generated | std | pearson | spearman |
| **MS CS** | **0** | GraphSage | 0.750 | 0.003 | 0.680 | 0.005 | 0.520 | 0.007 | 0.974 | 0.956 |
| | | AS-GCN | 0.090 | 0.027 | 0.070 | 0.035 | 0.070 | 0.016 | | |
| | | FastGCN | 0.920 | 0.001 | 0.910 | 0.001 | 0.820 | 0.001 | | |
| | | PASS | 0.870 | 0.007 | 0.810 | 0.008 | 0.640 | 0.015 | | |
| | **2** | GraphSage | 0.320 | 0.002 | 0.350 | 0.003 | 0.240 | 0.080 | | |
| | | AS-GCN | 0.040 | 0.028 | 0.050 | 0.022 | 0.050 | 0.036 | | |
| | | FastGCN | 0.910 | 0.002 | 0.910 | 0.001 | 0.820 | 0.002 | | |
| | | PASS | 0.810 | 0.005 | 0.750 | 0.003 | 0.660 | 0.004 | | |
| | **4** | GraphSage | 0.200 | 0.008 | 0.230 | 0.008 | 0.120 | 0.018 | | |
| | | AS-GCN | 0.070 | 0.033 | 0.050 | 0.027 | 0.040 | 0.038 | | |
| | | FastGCN | 0.900 | 0.005 | 0.890 | 0.003 | 0.610 | 0.007 | | |
| | | PASS | 0.790 | 0.013 | 0.730 | 0.005 | 0.500 | 0.011 | | |
| Dataset | #NE | model | Original | std | Cluster | std | Generated | std | pearson | spearman |
| **MS Physic** | **0** | GraphSage | 0.850 | 0.005 | 0.790 | 0.003 | 0.590 | 0.009 | 0.956 | 0.951 |
| | | AS-GCN | 0.240 | 0.051 | 0.190 | 0.042 | 0.240 | 0.052 | | |
| | | FastGCN | 0.950 | 0.001 | 0.940 | 0.001 | 0.820 | 0.004 | | |
| | | PASS | 0.920 | 0.000 | 0.860 | 0.003 | 0.670 | 0.006 | | |
| | **2** | GraphSage | 0.490 | 0.001 | 0.500 | 0.003 | 0.420 | 0.005 | | |
| | | AS-GCN | 0.160 | 0.022 | 0.210 | 0.032 | 0.130 | 0.055 | | |
| | | FastGCN | 0.940 | 0.004 | 0.930 | 0.005 | 0.800 | 0.009 | | |
| | | PASS | 0.900 | 0.009 | 0.840 | 0.008 | 0.690 | 0.012 | | |
| | **4** | GraphSage | 0.300 | 0.003 | 0.330 | 0.005 | 0.280 | 0.002 | | |
| | | AS-GCN | 0.340 | 0.005 | 0.090 | 0.052 | 0.080 | 0.039 | | |
| | | FastGCN | 0.930 | 0.002 | 0.920 | 0.003 | 0.780 | 0.001 | | |
| | | PASS | 0.890 | 0.001 | 0.830 | 0.005 | 0.610 | 0.004 | | |

Table 8: **GNN performance on SCENARIO 3: different sampling numbers on neighbor sampling.**

| Dataset | #SN | model | Original | std | Cluster | std | Generated | std | pearson | spearman |
|---|---|---|---|---|---|---|---|---|---|---|
| Cora | 0 | GraphSage | 0.750 | 0.013 | 0.560 | 0.028 | 0.500 | 0.011 | | |
| | | AS-GCN | 0.120 | 0.001 | 0.120 | 0.011 | 0.110 | 0.005 | | |
| | | FastGCN | 0.450 | 0.008 | 0.390 | 0.006 | 0.380 | 0.003 | | |
| | | PASS | 0.800 | 0.007 | 0.600 | 0.008 | 0.540 | 0.003 | | |
| | 2 | GraphSage | 0.830 | 0.007 | 0.740 | 0.008 | 0.690 | 0.018 | 0.967 | 0.814 |
| | | AS-GCN | 0.130 | 0.009 | 0.130 | 0.013 | 0.130 | 0.014 | | |
| | | FastGCN | 0.750 | 0.008 | 0.660 | 0.011 | 0.660 | 0.001 | | |
| | | PASS | 0.840 | 0.004 | 0.740 | 0.012 | 0.680 | 0.011 | | |
| | 4 | GraphSage | 0.850 | 0.001 | 0.810 | 0.004 | 0.600 | 0.005 | | |
| | | AS-GCN | 0.130 | 0.022 | 0.140 | 0.029 | 0.150 | 0.046 | | |
| | | FastGCN | 0.870 | 0.004 | 0.820 | 0.007 | 0.640 | 0.008 | | |
| | | PASS | 0.820 | 0.009 | 0.790 | 0.000 | 0.520 | 0.026 | | |
| **Dataset** | **#SN** | **model** | **Original** | **std** | **Cluster** | **std** | **Generated** | **std** | **pearson** | **spearman** |
| Citeseer | 0 | GraphSage | 0.680 | 0.014 | 0.500 | 0.011 | 0.440 | 0.016 | | |
| | | AS-GCN | 0.110 | 0.013 | 0.090 | 0.005 | 0.100 | 0.006 | | |
| | | FastGCN | 0.370 | 0.011 | 0.330 | 0.003 | 0.330 | 0.015 | | |
| | | PASS | 0.700 | 0.005 | 0.530 | 0.014 | 0.460 | 0.006 | | |
| | 2 | GraphSage | 0.710 | 0.004 | 0.610 | 0.006 | 0.560 | 0.003 | 0.973 | 0.904 |
| | | AS-GCN | 0.110 | 0.012 | 0.110 | 0.010 | 0.090 | 0.004 | | |
| | | FastGCN | 0.670 | 0.008 | 0.600 | 0.005 | 0.580 | 0.001 | | |
| | | PASS | 0.710 | 0.003 | 0.610 | 0.007 | 0.560 | 0.007 | | |
| | 4 | GraphSage | 0.730 | 0.006 | 0.650 | 0.009 | 0.600 | 0.01 | | |
| | | AS-GCN | 0.110 | 0.004 | 0.120 | 0.001 | 0.100 | 0.012 | | |
| | | FastGCN | 0.770 | 0.003 | 0.700 | 0.004 | 0.680 | 0.001 | | |
| | | PASS | 0.730 | 0.002 | 0.650 | 0.004 | 0.580 | 0.009 | | |
| **Dataset** | **#SN** | **model** | **Original** | **std** | **Cluster** | **std** | **Generated** | **std** | **pearson** | **spearman** |
| Pubmed | 1 | GraphSage | 0.780 | 0.003 | 0.680 | 0.005 | 0.600 | 0.004 | | |
| | | AS-GCN | 0.250 | 0.002 | 0.260 | 0.009 | 0.260 | 0.011 | | |
| | | FastGCN | 0.480 | 0.002 | 0.460 | 0.004 | 0.440 | 0.003 | | |
| | | PASS | 0.860 | 0.002 | 0.720 | 0.004 | 0.660 | 0.003 | | |
| | 3 | GraphSage | 0.830 | 0.003 | 0.780 | 0.005 | 0.710 | 0.001 | 0.989 | 0.824 |
| | | AS-GCN | 0.240 | 0.012 | 0.240 | 0.015 | 0.250 | 0.013 | | |
| | | FastGCN | 0.750 | 0.004 | 0.710 | 0.001 | 0.660 | 0.006 | | |
| | | PASS | 0.880 | 0.002 | 0.780 | 0.003 | 0.710 | 0.008 | | |
| | 5 | GraphSage | 0.850 | 0.001 | 0.800 | 0.001 | 0.740 | 0.002 | | |
| | | AS-GCN | 0.260 | 0.021 | 0.260 | 0.007 | 0.240 | 0.02 | | |
| | | FastGCN | 0.860 | 0.002 | 0.800 | 0.003 | 0.740 | 0.002 | | |
| | | PASS | 0.870 | 0.002 | 0.790 | 0.004 | 0.730 | 0.004 | | |
| **Dataset** | **#SN** | **model** | **Original** | **std** | **Cluster** | **std** | **Generated** | **std** | **pearson** | **spearman** |
| Amazon Computer | 1 | GraphSage | 0.670 | 0.010 | 0.550 | 0.008 | 0.450 | 0.01 | | |
| | | AS-GCN | 0.090 | 0.006 | 0.060 | 0.028 | 0.040 | 0.005 | | |
| | | FastGCN | 0.780 | 0.004 | 0.740 | 0.007 | 0.700 | 0.006 | | |
| | | PASS | 0.750 | 0.000 | 0.620 | 0.018 | 0.530 | 0.02 | | |
| | 3 | GraphSage | 0.790 | 0.003 | 0.700 | 0.015 | 0.600 | 0.015 | 0.975 | 0.890 |
| | | AS-GCN | 0.110 | 0.025 | 0.040 | 0.014 | 0.120 | 0.06 | | |
| | | FastGCN | 0.870 | 0.001 | 0.840 | 0.006 | 0.800 | 0.011 | | |
| | | PASS | 0.810 | 0.015 | 0.760 | 0.023 | 0.640 | 0.009 | | |
| | 5 | GraphSage | 0.770 | 0.008 | 0.720 | 0.004 | 0.680 | 0.005 | | |
| | | AS-GCN | 0.120 | 0.085 | 0.100 | 0.057 | 0.030 | 0.007 | | |
| | | FastGCN | 0.850 | 0.003 | 0.830 | 0.000 | 0.790 | 0.01 | | |
| | | PASS | 0.830 | 0.002 | 0.730 | 0.011 | 0.680 | 0.022 | | |
| **Dataset** | **#SN** | **model** | **Original** | **std** | **Cluster** | **std** | **Generated** | **std** | **pearson** | **spearman** |
| Amazon Photo | 1 | GraphSage | 0.740 | 0.016 | 0.660 | 0.003 | 0.500 | 0.014 | | |
| | | AS-GCN | 0.110 | 0.037 | 0.090 | 0.030 | 0.090 | 0.04 | | |
| | | FastGCN | 0.830 | 0.005 | 0.810 | 0.005 | 0.750 | 0.009 | | |
| | | PASS | 0.850 | 0.011 | 0.730 | 0.026 | 0.520 | 0.01 | | |
| | 3 | GraphSage | 0.840 | 0.006 | 0.810 | 0.007 | 0.740 | 0.007 | 0.961 | 0.931 |
| | | AS-GCN | 0.140 | 0.026 | 0.140 | 0.019 | 0.130 | 0.038 | | |
| | | FastGCN | 0.930 | 0.005 | 0.910 | 0.002 | 0.890 | 0.002 | | |
| | | PASS | 0.910 | 0.002 | 0.870 | 0.002 | 0.750 | 0.017 | | |
| | 5 | GraphSage | 0.910 | 0.010 | 0.890 | 0.002 | 0.780 | 0.009 | | |
| | | AS-GCN | 0.860 | 0.021 | 0.850 | 0.021 | 0.790 | 0.031 | | |
| | | FastGCN | 0.110 | 0.005 | 0.050 | 0.001 | 0.110 | 0.021 | | |
| | | PASS | 0.930 | 0.002 | 0.900 | 0.011 | 0.850 | 0.005 | | |
| **Dataset** | **#SN** | **model** | **Original** | **std** | **Cluster** | **std** | **Generated** | **std** | **pearson** | **spearman** |
| MS CS | 1 | GraphSage | 0.740 | 0.008 | 0.650 | 0.004 | 0.530 | 0.006 | | |
| | | AS-GCN | 0.070 | 0.050 | 0.060 | 0.025 | 0.080 | 0.023 | | |
| | | FastGCN | 0.920 | 0.001 | 0.920 | 0.000 | 0.840 | 0.003 | | |
| | | PASS | 0.870 | 0.005 | 0.770 | 0.005 | 0.690 | 0.004 | | |
| | 3 | GraphSage | 0.840 | 0.004 | 0.820 | 0.004 | 0.680 | 0.008 | 0.986 | 0.901 |
| | | AS-GCN | 0.090 | 0.051 | 0.090 | 0.035 | 0.070 | 0.018 | | |
| | | FastGCN | 0.930 | 0.001 | 0.920 | 0.002 | 0.810 | 0.01 | | |
| | | PASS | 0.900 | 0.004 | 0.870 | 0.003 | 0.680 | 0.013 | | |
| | 5 | GraphSage | 0.870 | 0.003 | 0.850 | 0.003 | 0.750 | 0.011 | | |
| | | AS-GCN | 0.060 | 0.044 | 0.040 | 0.002 | 0.110 | 0.037 | | |
| | | FastGCN | 0.930 | 0.001 | 0.920 | 0.000 | 0.810 | 0.01 | | |
| | | PASS | 0.910 | 0.001 | 0.880 | 0.001 | 0.710 | 0.014 | | |
| **Dataset** | **#SN** | **model** | **Original** | **std** | **Cluster** | **std** | **Generated** | **std** | **pearson** | **spearman** |
| MS Physic | 1 | GraphSage | 0.850 | 0.001 | 0.780 | 0.004 | 0.590 | 0.003 | | |
| | | AS-GCN | 0.240 | 0.125 | 0.260 | 0.139 | 0.140 | 0.021 | | |
| | | FastGCN | 0.950 | 0.001 | 0.940 | 0.001 | 0.840 | 0.002 | | |
| | | PASS | 0.920 | 0.003 | 0.850 | 0.004 | 0.650 | 0.004 | | |
| | 3 | GraphSage | 0.940 | 0.002 | 0.900 | 0.001 | 0.720 | 0.006 | 0.947 | 0.901 |
| | | AS-GCN | 0.910 | 0.001 | 0.880 | 0.002 | 0.730 | 0.022 | | |
| | | FastGCN | 0.390 | 0.025 | 0.210 | 0.033 | 0.230 | 0.034 | | |
| | | PASS | 0.950 | 0.003 | 0.940 | 0.002 | 0.820 | 0.009 | | |
| | 5 | GraphSage | 0.950 | 0.005 | 0.910 | 0.003 | 0.740 | 0.001 | | |
| | | AS-GCN | 0.930 | 0.001 | 0.900 | 0.001 | 0.760 | 0.001 | | |
| | | FastGCN | 0.090 | 0.036 | 0.150 | 0.048 | 0.260 | 0.033 | | |
| | | PASS | 0.960 | 0.002 | 0.940 | 0.003 | 0.830 | 0.020 | | |

Table 9: **GNN performance on SCENARIO 4: distribution shift.**

| Dataset | $\alpha$ | model | Original | std | Cluster | std | Generated | std | pearson | spearman |
|---|---|---|---|---|---|---|---|---|---|---|
| **Cora** | iid | GraphSage | 0.830 | 0.010 | 0.820 | 0.003 | 0.760 | 0.024 | 0.867 | 0.832 |
| | | SGC | 0.860 | 0.001 | 0.810 | 0.004 | 0.810 | 0.023 | | |
| | | GAT | 0.840 | 0.007 | 0.800 | 0.005 | 0.760 | 0.014 | | |
| | | PPNP | 0.840 | 0.007 | 0.800 | 0.008 | 0.810 | 0.016 | | |
| | 0.01 | GraphSage | 0.790 | 0.007 | 0.780 | 0.010 | 0.650 | 0.011 | | |
| | | SGC | 0.820 | 0.003 | 0.780 | 0.002 | 0.710 | 0.001 | | |
| | | GAT | 0.780 | 0.007 | 0.760 | 0.005 | 0.680 | 0.005 | | |
| | | PPNP | 0.780 | 0.005 | 0.760 | 0.004 | 0.730 | 0.001 | | |
| | 0.3 | GraphSage | 0.730 | 0.010 | 0.730 | 0.003 | 0.660 | 0.012 | | |
| | | SGC | 0.790 | 0.003 | 0.720 | 0.002 | 0.700 | 0.01 | | |
| | | GAT | 0.760 | 0.003 | 0.700 | 0.004 | 0.650 | 0.016 | | |
| | | PPNP | 0.770 | 0.008 | 0.730 | 0.006 | 0.680 | 0.017 | | |
| Dataset | $\alpha$ | model | Original | std | Cluster | std | Generated | std | pearson | spearman |
| **Citeseer** | iid | GraphSage | 0.690 | 0.005 | 0.640 | 0.003 | 0.570 | 0.021 | 0.812 | 0.799 |
| | | SGC | 0.710 | 0.004 | 0.650 | 0.001 | 0.590 | 0.017 | | |
| | | GAT | 0.680 | 0.016 | 0.650 | 0.003 | 0.580 | 0.011 | | |
| | | PPNP | 0.690 | 0.002 | 0.630 | 0.002 | 0.610 | 0.007 | | |
| | 0.01 | GraphSage | 0.590 | 0.009 | 0.550 | 0.012 | 0.510 | 0.018 | | |
| | | SGC | 0.640 | 0.002 | 0.580 | 0.003 | 0.560 | 0.014 | | |
| | | GAT | 0.610 | 0.005 | 0.550 | 0.003 | 0.510 | 0.022 | | |
| | | PPNP | 0.610 | 0.010 | 0.550 | 0.010 | 0.540 | 0.02 | | |
| | 0.3 | GraphSage | 0.610 | 0.006 | 0.580 | 0.002 | 0.500 | 0.02 | | |
| | | SGC | 0.660 | 0.003 | 0.560 | 0.002 | 0.530 | 0.012 | | |
| | | GAT | 0.650 | 0.007 | 0.560 | 0.005 | 0.510 | 0.003 | | |
| | | PPNP | 0.630 | 0.001 | 0.550 | 0.005 | 0.550 | 0.012 | | |
| Dataset | $\alpha$ | model | Original | std | Cluster | std | Generated | std | pearson | spearman |
| **Pubmed** | iid | GraphSage | 0.840 | 0.002 | 0.810 | 0.002 | 0.720 | 0.009 | 0.830 | 0.794 |
| | | SGC | 0.860 | 0.001 | 0.820 | 0.000 | 0.730 | 0.005 | | |
| | | GAT | 0.840 | 0.005 | 0.810 | 0.002 | 0.720 | 0.014 | | |
| | | PPNP | 0.820 | 0.002 | 0.800 | 0.002 | 0.730 | 0.004 | | |
| | 0.01 | GraphSage | 0.810 | 0.007 | 0.750 | 0.008 | 0.660 | 0.01 | | |
| | | SGC | 0.800 | 0.002 | 0.760 | 0.004 | 0.680 | 0.007 | | |
| | | GAT | 0.790 | 0.005 | 0.760 | 0.005 | 0.660 | 0.021 | | |
| | | PPNP | 0.770 | 0.004 | 0.760 | 0.006 | 0.680 | 0.008 | | |
| | 0.3 | GraphSage | 0.770 | 0.007 | 0.720 | 0.005 | 0.620 | 0.014 | | |
| | | SGC | 0.770 | 0.003 | 0.730 | 0.000 | 0.660 | 0.003 | | |
| | | GAT | 0.750 | 0.014 | 0.700 | 0.002 | 0.630 | 0.008 | | |
| | | PPNP | 0.740 | 0.009 | 0.730 | 0.004 | 0.660 | 0.001 | | |
| Dataset | $\alpha$ | model | Original | std | Cluster | std | Generated | std | pearson | spearman |
| **Amazon Computer** | iid | GraphSage | 0.850 | 0.009 | 0.800 | 0.012 | 0.790 | 0.008 | 0.906 | 0.860 |
| | | SGC | 0.870 | 0.004 | 0.790 | 0.004 | 0.800 | 0.003 | | |
| | | GAT | 0.840 | 0.003 | 0.790 | 0.008 | 0.800 | 0.012 | | |
| | | PPNP | 0.840 | 0.003 | 0.800 | 0.005 | 0.810 | 0.003 | | |
| | 0.01 | GraphSage | 0.790 | 0.013 | 0.740 | 0.010 | 0.750 | 0.003 | | |
| | | SGC | 0.800 | 0.003 | 0.750 | 0.006 | 0.740 | 0.003 | | |
| | | GAT | 0.770 | 0.028 | 0.750 | 0.005 | 0.750 | 0.006 | | |
| | | PPNP | 0.770 | 0.015 | 0.750 | 0.003 | 0.760 | 0.007 | | |
| | 0.3 | GraphSage | 0.750 | 0.020 | 0.710 | 0.015 | 0.690 | 0.019 | | |
| | | SGC | 0.760 | 0.004 | 0.710 | 0.005 | 0.710 | 0.006 | | |
| | | GAT | 0.760 | 0.003 | 0.720 | 0.010 | 0.700 | 0.006 | | |
| | | PPNP | 0.740 | 0.004 | 0.710 | 0.009 | 0.710 | 0.021 | | |
| Dataset | $\alpha$ | model | Original | std | Cluster | std | Generated | std | pearson | spearman |
| **Amazon Photo** | iid | GraphSage | 0.890 | 0.001 | 0.890 | 0.002 | 0.910 | 0.003 | 0.771 | 0.847 |
| | | SGC | 0.890 | 0.005 | 0.890 | 0.002 | 0.911 | 0.007 | | |
| | | GAT | 0.880 | 0.002 | 0.870 | 0.008 | 0.910 | 0.003 | | |
| | | PPNP | 0.880 | 0.002 | 0.900 | 0.002 | 0.910 | 0.006 | | |
| | 0.01 | GraphSage | 0.880 | 0.014 | 0.850 | 0.016 | 0.850 | 0.012 | | |
| | | SGC | 0.880 | 0.008 | 0.860 | 0.006 | 0.840 | 0.015 | | |
| | | GAT | 0.860 | 0.011 | 0.850 | 0.002 | 0.830 | 0.007 | | |
| | | PPNP | 0.860 | 0.009 | 0.860 | 0.003 | 0.850 | 0.019 | | |
| | 0.3 | GraphSage | 0.830 | 0.011 | 0.860 | 0.018 | 0.830 | 0.009 | | |
| | | SGC | 0.850 | 0.013 | 0.820 | 0.002 | 0.790 | 0.017 | | |
| | | GAT | 0.840 | 0.015 | 0.850 | 0.027 | 0.820 | 0.006 | | |
| | | PPNP | 0.860 | 0.015 | 0.860 | 0.007 | 0.850 | 0.02 | | |
| Dataset | $\alpha$ | model | Original | std | Cluster | std | Generated | std | pearson | spearman |
| **MS CS** | iid | GraphSage | 0.870 | 0.004 | 0.880 | 0.001 | 0.850 | 0.011 | 0.792 | 0.751 |
| | | SGC | 0.870 | 0.006 | 0.880 | 0.002 | 0.850 | 0.012 | | |
| | | GAT | 0.869 | 0.001 | 0.860 | 0.003 | 0.830 | 0.007 | | |
| | | PPNP | 0.870 | 0.006 | 0.880 | 0.002 | 0.840 | 0.008 | | |
| | 0.01 | GraphSage | 0.800 | 0.003 | 0.820 | 0.012 | 0.790 | 0.006 | | |
| | | SGC | 0.880 | 0.002 | 0.860 | 0.002 | 0.830 | 0.003 | | |
| | | GAT | 0.850 | 0.004 | 0.840 | 0.006 | 0.800 | 0.01 | | |
| | | PPNP | 0.840 | 0.003 | 0.860 | 0.001 | 0.830 | 0.003 | | |
| | 0.3 | GraphSage | 0.820 | 0.008 | 0.850 | 0.007 | 0.800 | 0.005 | | |
| | | SGC | 0.870 | 0.002 | 0.850 | 0.001 | 0.840 | 0.003 | | |
| | | GAT | 0.850 | 0.008 | 0.840 | 0.003 | 0.810 | 0.006 | | |
| | | PPNP | 0.840 | 0.001 | 0.850 | 0.003 | 0.830 | 0.005 | | |
| Dataset | $\alpha$ | model | Original | std | Cluster | std | Generated | std | pearson | spearman |
| **MS Physic** | iid | GraphSage | 0.930 | 0.002 | 0.930 | 0.002 | 0.840 | 0.008 | 0.925 | 0.815 |
| | | SGC | 0.920 | 0.001 | 0.920 | 0.001 | 0.840 | 0.007 | | |
| | | GAT | 0.930 | 0.002 | 0.920 | 0.002 | 0.820 | 0.011 | | |
| | | PPNP | 0.930 | 0.005 | 0.930 | 0.000 | 0.840 | 0.007 | | |
| | 0.01 | GraphSage | 0.830 | 0.033 | 0.850 | 0.004 | 0.760 | 0.019 | | |
| | | SGC | 0.840 | 0.004 | 0.820 | 0.005 | 0.740 | 0.015 | | |
| | | GAT | 0.870 | 0.007 | 0.840 | 0.011 | 0.780 | 0.009 | | |
| | | PPNP | 0.840 | 0.007 | 0.830 | 0.006 | 0.740 | 0.009 | | |
| | 0.3 | GraphSage | 0.840 | 0.012 | 0.840 | 0.009 | 0.680 | 0.023 | | |
| | | SGC | 0.810 | 0.009 | 0.820 | 0.003 | 0.700 | 0.009 | | |
| | | GAT | 0.850 | 0.011 | 0.840 | 0.002 | 0.720 | 0.019 | | |
| | | PPNP | 0.810 | 0.012 | 0.830 | 0.004 | 0.700 | 0.009 | | |

Table 10: **GNN performance on link prediction.**

| Dataset | predictor | model | Original | std | Cluster | std | Generated | std | pearson | spearman |
|---|---|---|---|---|---|---|---|---|---|---|
| Cora | Dot | GCN | 0.720 | 0.010 | 0.770 | 0.009 | 0.680 | 0.012 | 0.781 | 0.741 |
| | | SGC | 0.710 | 0.025 | 0.760 | 0.005 | 0.660 | 0.016 | | |
| | | GIN | 0.820 | 0.015 | 0.760 | 0.016 | 0.650 | 0.022 | | |
| | | GAT | 0.810 | 0.002 | 0.810 | 0.007 | 0.730 | 0.015 | | |
| | MLP | GCN | 0.540 | 0.005 | 0.620 | 0.012 | 0.510 | 0.01 | | |
| | | SGC | 0.530 | 0.016 | 0.590 | 0.042 | 0.510 | 0.006 | | |
| | | GIN | 0.530 | 0.012 | 0.690 | 0.016 | 0.630 | 0.017 | | |
| | | GAT | 0.550 | 0.003 | 0.660 | 0.013 | 0.610 | 0.034 | | |
| Dataset | predictor | model | Original | std | Cluster | std | Generated | std | pearson | spearman |
| Citeseer | Dot | GCN | 0.690 | 0.007 | 0.740 | 0.009 | 0.650 | 0.026 | 0.808 | 0.824 |
| | | SGC | 0.700 | 0.003 | 0.730 | 0.013 | 0.670 | 0.022 | | |
| | | GIN | 0.830 | 0.008 | 0.720 | 0.003 | 0.650 | 0.01 | | |
| | | GAT | 0.750 | 0.005 | 0.780 | 0.012 | 0.680 | 0.021 | | |
| | MLP | GCN | 0.580 | 0.005 | 0.650 | 0.012 | 0.590 | 0.01 | | |
| | | SGC | 0.580 | 0.008 | 0.640 | 0.025 | 0.590 | 0.023 | | |
| | | GIN | 0.570 | 0.011 | 0.720 | 0.012 | 0.610 | 0.024 | | |
| | | GAT | 0.610 | 0.005 | 0.680 | 0.001 | 0.620 | 0.009 | | |
| Dataset | predictor | model | Original | std | Cluster | std | Generated | std | pearson | spearman |
| Pubmed | Dot | GCN | 0.800 | 0.018 | 0.810 | 0.005 | 0.670 | 0.019 | 0.725 | 0.420 |
| | | SGC | 0.790 | 0.002 | 0.780 | 0.006 | 0.660 | 0.004 | | |
| | | GIN | 0.800 | 0.008 | 0.760 | 0.008 | 0.650 | 0.009 | | |
| | | GAT | 0.860 | 0.003 | 0.850 | 0.007 | 0.720 | 0.008 | | |
| | MLP | GCN | 0.760 | 0.003 | 0.770 | 0.012 | 0.640 | 0.017 | | |
| | | SGC | 0.770 | 0.006 | 0.770 | 0.006 | 0.610 | 0.008 | | |
| | | GIN | 0.750 | 0.004 | 0.790 | 0.014 | 0.660 | 0.004 | | |
| | | GAT | 0.750 | 0.004 | 0.850 | 0.019 | 0.660 | 0.011 | | |
| Dataset | predictor | model | Original | std | Cluster | std | Generated | std | pearson | spearman |
| Amazon Computer | Dot | GCN | 0.790 | 0.010 | 0.850 | 0.026 | 0.810 | 0.008 | 0.652 | 0.559 |
| | | SGC | 0.760 | 0.005 | 0.770 | 0.030 | 0.730 | 0.025 | | |
| | | GIN | 0.800 | 0.013 | 0.880 | 0.004 | 0.830 | 0.005 | | |
| | | GAT | 0.750 | 0.057 | 0.840 | 0.014 | 0.560 | 0.08 | | |
| | MLP | GCN | 0.810 | 0.005 | 0.890 | 0.005 | 0.830 | 0.012 | | |
| | | SGC | 0.800 | 0.000 | 0.850 | 0.020 | 0.730 | 0.021 | | |
| | | GIN | 0.800 | 0.003 | 0.890 | 0.010 | 0.810 | 0.01 | | |
| | | GAT | 0.860 | 0.005 | 0.910 | 0.005 | 0.800 | 0.005 | | |
| Dataset | predictor | model | Original | std | Cluster | std | Generated | std | pearson | spearman |
| Amazon Photo | Dot | GCN | 0.890 | 0.011 | 0.920 | 0.005 | 0.860 | 0.016 | 0.887 | 0.443 |
| | | SGC | 0.810 | 0.014 | 0.840 | 0.015 | 0.780 | 0.011 | | |
| | | GIN | 0.810 | 0.007 | 0.910 | 0.006 | 0.880 | 0.002 | | |
| | | GAT | 0.530 | 0.023 | 0.740 | 0.151 | 0.660 | 0.134 | | |
| | MLP | GCN | 0.870 | 0.006 | 0.930 | 0.006 | 0.890 | 0.001 | | |
| | | SGC | 0.840 | 0.010 | 0.900 | 0.012 | 0.810 | 0.015 | | |
| | | GIN | 0.850 | 0.006 | 0.930 | 0.002 | 0.870 | 0.004 | | |
| | | GAT | 0.910 | 0.007 | 0.930 | 0.004 | 0.850 | 0.007 | | |

Table 11: **Benchmark effectiveness across 9 GNN models without any graph variations**

| Dataset | GNN model | Original | Generated | Pearson | Spearman |
| --- | --- | --- | --- | --- | --- |
| Cora | GCN | 0.860 | 0.760 | | |
| | SGC | 0.850 | 0.750 | | |
| | GIN | 0.850 | 0.750 | | |
| | GAT | 0.830 | 0.750 | | |
| | GraphSage | 0.750 | 0.500 | 0.939 | 0.868 |
| | AS-GCN | 0.120 | 0.110 | | |
| | FastGCN | 0.450 | 0.380 | | |
| | PASS | 0.800 | 0.540 | | |
| | PPNP | 0.840 | 0.810 | | |
| Dataset | GNN model | Original | Generated | Pearson | Spearman |
| Citeseer | GCN | 0.730 | 0.590 | | |
| | SGC | 0.730 | 0.580 | | |
| | GIN | 0.710 | 0.570 | | |
| | GAT | 0.710 | 0.570 | | |
| | GraphSage | 0.680 | 0.440 | 0.948 | 0.743 |
| | AS-GCN | 0.110 | 0.100 | | |
| | FastGCN | 0.370 | 0.330 | | |
| | PASS | 0.700 | 0.460 | | |
| | PPNP | 0.690 | 0.610 | | |
| Dataset | GNN model | Original | Generated | Pearson | Spearman |
| Pubmed | GCN | 0.860 | 0.780 | | |
| | SGC | 0.860 | 0.780 | | |
| | GIN | 0.830 | 0.770 | | |
| | GAT | 0.860 | 0.780 | | |
| | GraphSage | 0.780 | 0.600 | 0.962 | 0.868 |
| | AS-GCN | 0.250 | 0.260 | | |
| | FastGCN | 0.480 | 0.440 | | |
| | PASS | 0.860 | 0.660 | | |
| | PPNP | 0.820 | 0.730 | | |
| Dataset | GNN model | Original | Generated | Pearson | Spearman |
| Amazon Computer | GCN | 0.860 | 0.840 | | |
| | SGC | 0.860 | 0.830 | | |
| | GIN | 0.850 | 0.800 | | |
| | GAT | 0.840 | 0.830 | | |
| | GraphSage | 0.670 | 0.450 | 0.952 | 0.920 |
| | AS-GCN | 0.090 | 0.040 | | |
| | FastGCN | 0.780 | 0.700 | | |
| | PASS | 0.750 | 0.530 | | |
| | PPNP | 0.840 | 0.810 | | |
| Dataset | GNN model | Original | Generated | Pearson | Spearman |
| Amazon Photo | GCN | 0.910 | 0.900 | | |
| | SGC | 0.910 | 0.900 | | |
| | GIN | 0.900 | 0.900 | | |
| | GAT | 0.900 | 0.890 | | |
| | GraphSage | 0.740 | 0.500 | 0.899 | 0.786 |
| | AS-GCN | 0.110 | 0.090 | | |
| | FastGCN | 0.830 | 0.750 | | |
| | PASS | 0.850 | 0.520 | | |
| | PPNP | 0.880 | 0.910 | | |
| Dataset | GNN model | Original | Generated | Pearson | Spearman |
| MS CS | GCN | 0.880 | 0.830 | | |
| | SGC | 0.880 | 0.830 | | |
| | GIN | 0.870 | 0.820 | | |
| | GAT | 0.880 | 0.830 | | |
| | GraphSage | 0.740 | 0.530 | 0.964 | 0.784 |
| | AS-GCN | 0.070 | 0.080 | | |
| | FastGCN | 0.920 | 0.840 | | |
| | PASS | 0.870 | 0.690 | | |
| | PPNP | 0.870 | 0.840 | | |
| Dataset | GNN model | Original | Generated | Pearson | Spearman |
| MS Physic | GCN | 0.930 | 0.840 | | |
| | SGC | 0.920 | 0.840 | | |
| | GIN | 0.930 | 0.820 | | |
| | GAT | 0.930 | 0.840 | | |
| | GraphSage | 0.850 | 0.590 | 0.947 | 0.776 |
| | AS-GCN | 0.240 | 0.140 | | |
| | FastGCN | 0.950 | 0.840 | | |
| | PASS | 0.920 | 0.650 | | |
| | PPNP | 0.930 | 0.840 | | |

## A.5    ADDITIONAL GRAPH STATISTICS

Figure 5 shows distributions of graph statistics on computation graphs sampled from the original/quantized/generated graphs. Quantized graphs are graphs after the quantization process: each feature vector is replaced by the mean feature vector of a cluster it belongs to, and adjacency matrices are a constant encoded by the duplicate encoding scheme. Quantized graphs are input to CGT, and generated graphs are output from CGT as presented in Figure 2. While converting from original graphs to quantized graphs, CGT trades off some of the graph statistics information for k-anonymity

Table 12: **Privacy-Performance trade-off in graph generation on the Cora dataset**

| #NE | model | Original | No privacy | K-anonymity | | | DP kmean ($\delta = 0.01$) | | | DP SGD ($\delta = 0.1$) | |
|---|---|---|---|---|---|---|---|---|---|---|---|
| | | | | $k = 100$ | $k = 500$ | $k = 1000$ | $\epsilon = 1$ | $\epsilon = 10$ | $\epsilon = 25$ | $\epsilon = 10^6$ | $\epsilon = 10^9$ |
| 0 | GCN | 0.860 | 0.760 | 0.750 | 0.520 | 0.120 | 0.530 | 0.570 | 0.650 | 0.130 | 0.640 |
| | SGC | 0.850 | 0.750 | 0.740 | 0.490 | 0.120 | 0.510 | 0.590 | 0.620 | 0.150 | 0.620 |
| | GIN | 0.850 | 0.750 | 0.760 | 0.510 | 0.110 | 0.520 | 0.570 | 0.650 | 0.140 | 0.640 |
| | GAT | 0.830 | 0.750 | 0.740 | 0.520 | 0.080 | 0.440 | 0.560 | 0.640 | 0.140 | 0.610 |
| 2 | GCN | 0.770 | 0.680 | 0.570 | 0.380 | 0.110 | 0.500 | 0.400 | 0.450 | 0.110 | 0.580 |
| | SGC | 0.770 | 0.680 | 0.580 | 0.360 | 0.080 | 0.350 | 0.410 | 0.450 | 0.140 | 0.570 |
| | GIN | 0.780 | 0.670 | 0.590 | 0.390 | 0.140 | 0.390 | 0.410 | 0.470 | 0.140 | 0.580 |
| | GAT | 0.680 | 0.660 | 0.560 | 0.380 | 0.110 | 0.350 | 0.390 | 0.430 | 0.120 | 0.530 |
| 4 | GCN | 0.720 | 0.610 | 0.510 | 0.280 | 0.090 | 0.280 | 0.390 | 0.430 | 0.100 | 0.410 |
| | SGC | 0.720 | 0.600 | 0.500 | 0.280 | 0.110 | 0.300 | 0.410 | 0.450 | 0.140 | 0.410 |
| | GIN | 0.660 | 0.590 | 0.480 | 0.300 | 0.160 | 0.320 | 0.410 | 0.460 | 0.150 | 0.400 |
| | GAT | 0.600 | 0.570 | 0.470 | 0.290 | 0.080 | 0.250 | 0.370 | 0.450 | 0.140 | 0.380 |
| Pearson | | 1.000 | 0.934 | 0.916 | 0.862 | 0.030 | 0.874 | 0.844 | 0.804 | 0.112 | 0.890 |
| Spearman | | 1.000 | 0.935 | 0.947 | 0.812 | 0.018 | 0.869 | 0.805 | 0.807 | 0.116 | 0.959 |

privacy benefits. In Figure 5, we can see distributions of graphs statistics have changed slightly from original graphs to quantized graphs. Then CGT learns distributions of graph statistics on the quantized graphs and generates synthetic graphs. The variations given by CGT are presented as differences in distributions between quantized and generated graphs in Figure 5.

## A.6 ABLATION STUDY

To show the importance of each component in our Computation Graph Transformer, we run three ablation studies on our model. Table 13 shows CGT without label conditioning (conditioning on the label of the root node of the computation graph), positional embedding trick (giving the same positional embedding to nodes at the same layers on the computation graph), and masked attention trick (attended only on direct ancestor nodes on the computation graph), respectively. When we remove the positional embedding trick, we provide the different positional embeddings to all nodes in a computation graph, following the original transformer architecture. When we remove attention masks from our model, the transformer attends all other nodes in the computation graphs to compute the context embeddings. As shown in Table 13, removing any component negatively impacts model performance. This shows not only the importance of label conditioning and our designed positional embeddings and attention masks, but also tricky aspects of graph generation and relations among graph structure, node attributes and labels.

## A.7 GRAPH NEURAL NETWORKS

We briefly review graph neural networks (GNNs) then describe how neighbor sampling operations can be applied on GNNs.

**Notations.** Let $\mathcal{G} = (\mathcal{V}, \mathcal{E})$ denote a graph with $n$ nodes $v_i \in \mathcal{V}$ and edges $(v_i, v_j) \in \mathcal{E}$. Denote an adjacency matrix $A = (a(v_i, v_j)) \in \mathbb{R}^{n \times n}$ and a feature matrix $X \in \mathbb{R}^{n \times d}$ where $x_i$ denotes the $d$-dimensional feature vector of node $v_i$.

**GCN (Kipf & Welling, 2016a).** GCN models stack layers of first-order spectral filters followed by a nonlinear activation functions to learn node embeddings. When $h_i^{(l)}$ denotes the hidden embeddings of node $v_i$ in the $l$-th layer, the simple and general form of GCNs is as follows:

$$h_i^{(l+1)} = \alpha(\frac{1}{n(i)} \sum_{j=1}^{n} a(v_i, v_j) h_j^{(l)} W^{(l)}), \quad l = 0, \dots, L-1 \tag{1}$$

where $a(v_i, v_j)$ is set to 1 when there is an edge from $v_i$ to $v_j$, otherwise 0. $n(i) = \sum_{j=1}^{n} a(v_i, v_j)$ is the degree of node $v_i$; $\alpha(\cdot)$ is a nonlinear function; $W^{(l)} \in \mathbb{R}^{d^{(l)} \times d^{(l+1)}}$ is the learnable transformation matrix in the $l$-th layer with $d^{(l)}$ denoting the hidden dimension at the $l$-th layer. $h_i^{(0)}$ is set with the input node attribute $x_i$

**GraphSage (Hamilton et al., 2017).** GCNs require the full expansion of neighborhoods across layers, leading to high computation and memory costs. To circumvent this issue, GraphSage adds sampling operations to GCNs to regulate the size of neighborhood. We first recast Equation 1 as

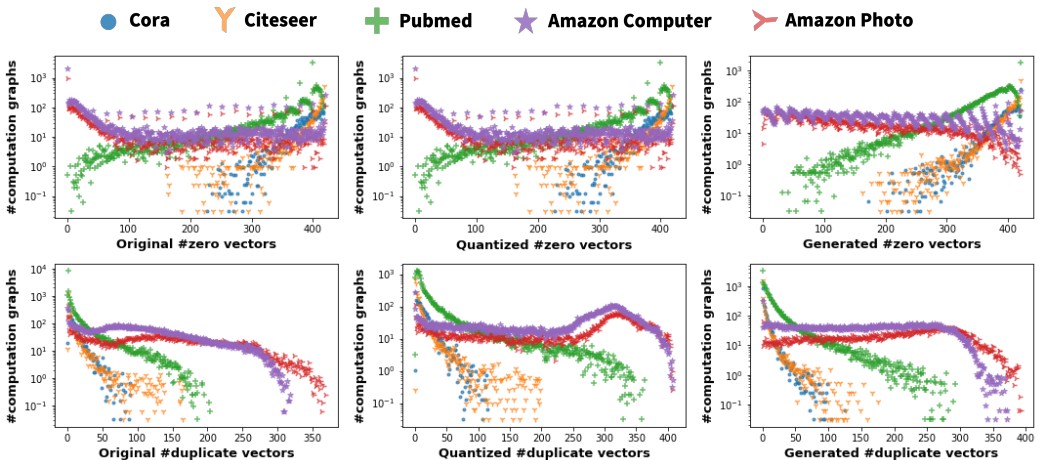

Figure 5: **CGT preserves distributions of graph statistics in generated graphs for each dataset:** While converting from original graphs to quantized graphs, CGT loses some of graph statistics information for k-anonymity privacy benefit. The variations given by CGT are presented as differences in distributions between quantized and generated graphs

Table 13: **Ablation study.** *Label*, *Position*, and *Attention* denote ablation of label conditioning, positional embeddings, and masked attention proposed in Section 4.1, respectively. Pearson scores measure the correlation in ranking of GNN models on original and generated graphs; a score of 1 denotes perfect correlation.

| Dataset | Model | Original | Label | Position | Attention | Ours |
|---------|-------|----------|-------|----------|-----------|------|
| Cora | GCN | 0.860 | 0.510 | 0.710 | 0.580 | 0.760 |
| | SGC | 0.850 | 0.520 | 0.700 | 0.580 | 0.750 |
| | GIN | 0.850 | 0.510 | 0.620 | 0.600 | 0.750 |
| | GAT | 0.830 | 0.520 | 0.450 | 0.350 | 0.750 |
| Citeseer | GCN | 0.730 | 0.450 | 0.670 | 0.530 | 0.590 |
| | SGC | 0.730 | 0.460 | 0.640 | 0.530 | 0.580 |
| | GIN | 0.710 | 0.450 | 0.520 | 0.530 | 0.570 |
| | GAT | 0.710 | 0.460 | 0.210 | 0.590 | 0.570 |
| Pubmed | GCN | 0.860 | 0.680 | 0.970 | 0.670 | 0.780 |
| | SGC | 0.860 | 0.680 | 0.970 | 0.580 | 0.780 |
| | GIN | 0.830 | 0.670 | 0.990 | 0.670 | 0.770 |
| | GAT | 0.860 | 0.690 | 0.940 | 0.120 | 0.780 |
| **Pearson** | | 1.000 | 0.718 | 0.647 | -0.097 | 0.988 |
| **Spearman** | | 1.000 | 0.799 | 0.610 | 0.149 | 0.911 |

follows:

$$h_i^{(l+1)} = \alpha_{W^{(l)}}(\mathbb{E}_{j\sim p(j|i)}[h_j^{(l)}]), \quad l = 0, \ldots, L-1 \tag{2}$$

where we combine the transformation matrix $W^{(l)}$ into the activation function $\alpha_{W^{(l)}}(\cdot)$ for concision; $p(j|i) = \frac{a(v_i,v_j)}{n(i)}$ defines the probability of sampling $v_j$ given $v_i$. Then we approximate the expectation by Monte-Carlo sampling as follows:

$$h_i^{(l+1)} = \alpha_{W^{(l)}}\left(\frac{1}{s}\sum_{j\sim p(j|i)}^{s} h_j^{(l)}\right), \quad l = 0, \ldots, L-1 \tag{3}$$

where $s$ is the number of sampled neighbors for each node. Now, we can regulate the size of neighborhood using $s$, in other words, the computational footprint for each minibatch.

### A.7.1 GNN MODELS USED IN THE BENCHMARK EFFECTIVENESS EXPERIMENT IN SECTION 5.3

We choose four different GNN models with different aggregation strategies to examine the effect of noisy edges on the aggregation strategies: GCN (Kipf & Welling, 2016a) with mean aggregator, GIN (Xu et al., 2018) with sum aggregator, SGC (Wu et al., 2019) with linear aggregator, and GAT (Veličković et al., 2017) with attention aggregator. We choose four different GNN models with different neighbor sampling strategies to examine the effect of noisy edges and number of

sampled neighbor numbers on GNN performance: GraphSage (Hamilton et al., 2017) with random sampling, FastGCN (Chen et al., 2018) with heuristic layer-wise sampling, AS-GCN (Huang et al., 2018) with trainable layer-wise sampling, and PASS (Yoon et al., 2021) with trainable node-wise sampling. Finally, we choose four different GNN models to check their robustness to distribution shifts in training/test time, as the authors of the original paper (Zhu et al., 2021) chose for their baselines: GCN (Kipf & Welling, 2016a), SGC (Wu et al., 2019), GAT (Veličković et al., 2017), and PPNP (Klicpera et al., 2018).

We implement GCN, SGC, GIN, and GAT from scratch for the *SCENARIO 1: noisy edges on aggregation strategies*. For *SCENARIOS 2 and 3: noisy edges and different sampling numbers on neighbor sampling*, we use open source implementations of each GNN model, ASGCN [1], FastGCN [2], and PASS [3], uploaded by the original authors. Finally, for *SCENARIO 4: distribution shift*, we use GCN, SGC, GAT, and PPNP implemented by (Zhu et al., 2021) using DGL library [4].

## A.8    ARCHITECTURE OF COMPUTATION GRAPH TRANSFORMER

Given a sequence $\mathbf{s} = [s_1, \cdots, s_T]$, the $M$-layered transformer maximizes the likelihood under the forward auto-regressive factorization as follow:

$$\max_\theta \log p_\theta(\mathbf{s}) = \sum_{t=1}^{T} \log p_\theta(s_t | \mathbf{s}_{<t}) = \sum_{t=1}^{T} \log \frac{exp(q_\theta^{(L)}(\mathbf{s}_{1:t-1})^\top e(s_t))}{\sum_{s' \neq s_t} exp(q_\theta^{(L)}(\mathbf{s}_{1:t-1})^\top e(s'))}$$

where node embedding $e(s_t)$ maps discrete input id $s_t$ to a randomly initialized trainable vector, and query embedding $q_\theta^{(L)}(\mathbf{s}_{1:t-1})$ encodes information until $(t-1)$-th token in the sequence. Query embedding $q_t^{(l)}$ is computed with context embeddings $\mathbf{h}_{1:t-1}^{(l-1)}$ of previous $t-1$ tokens and query embedding $q_t^{(l-1)}$ from the previous layer. Context embedding $h_t^{(l)}$ is computed from $\mathbf{h}_{1:t}^{(l-1)}$, context embeddings of previous $t-1$ tokens and $t$-th token from the previous layer. Note that, while the query embeddings have access only to the previous context embeddings $\mathbf{h}_{1:t-1}^{(l)}$, the context embeddings attend to all tokens $\mathbf{h}_{1:t}^{(l)}$. The context embedding $h_t^{(0)}$ is initially encoded by node embeddings $e(s_t)$ and position embedding $p_{l(t)}$ that encodes the location of each token in the sequence. The query embedding is initialized with a trainable vector and label embeddings $y_{s_1}$ as shown in Figure 3. This two streams (query and context) of self-attention layers are stacked $M$ time and predict the next tokens auto-regressively.

## A.9    DIFFERENTIALLY PRIVATE k-MEANS AND SGD ALGORITHMS

Given a set of data points, k-means clustering identifies k points, called cluster centers, by minimize the sum of distances of the data points from their closest cluster center. However, releasing the set of cluster centers could potentially leak information about particular users. For instance, if a particular data point is significantly far from the rest of the points, so the k-means clustering algorithm returns this single point as a cluster center. Then sensitive information about this single point could be revealed. To address this, DP k-means clustering algorithm (Chang et al., 2021) is designed within the framework of differential privacy. To generate the private core-set, DP k-means partitions the points into buckets of similar points then replaces each bucket by a single weighted point, while adding noise to both the counts and averages of points within a bucket.

Training a model is done through access to its parameter gradients, i.e., the gradients of the loss with respect to each parameter of the model. If this access preserves differential privacy of the training data, so does the resulting model, per the post-processing property of differential privacy. To achieve this goal, DP stochastic gradient descent (DP-SGD) (Song et al., 2013) modifies the minibatch stochastic optimization process to make it differentially private.

---

[1]`https://github.com/huangwb/AS-GCN`
[2]`https://github.com/matenure/FastGCN`
[3]`https://github.com/linkedin/PASS-GNN`
[4]`https://github.com/GentleZhu/Shift-Robust-GNNs`

Table 14: **Dataset statistics.**

| Dataset | Nodes | Edges | Features | Labels |
|---|---|---|---|---|
| Cora | 2,485 | 5,069 | 1,433 | 7 |
| Citeseer | 2,110 | 3,668 | 3,703 | 6 |
| Pubmed | 19,717 | 44,324 | 500 | 3 |
| Amazon Computer | 13,381 | 245,778 | 767 | 10 |
| Amazon Photo | 7,487 | 119,043 | 745 | 8 |
| MS CS | 18,333 | 81,894 | 6,805 | 15 |
| MS Physic | 34,493 | 247,962 | 8,415 | 5 |

We use the open source implementation of DP k-means provided by Google's differential privacy libraries [5]. We extend implementations of DP SGD provided by a public differential library Opacus [6].

## A.10 EXPERIMENTAL SETTINGS

All experiments were conducted on the same p3.2xlarge Amazon EC2 instance. We run each experiment three times and report the mean and standard deviation.

We evaluate on seven public datasets — three citation networks (Cora, Citeseer, and Pubmed) (Sen et al., 2008), two co-purchase graphs (Amazon Computer and Amazon Photo) (Shchur et al., 2018), and two co-authorship graph (MS CS and MS Physic) (Shchur et al., 2018). We use all nodes when training CGT. For GNN training, we split 50%/10%/40% of each dataset into the training/validation/test sets, respectively. We report their statistics in Table 14.

For the molecule graph generative models, GraphAF, GraphDF, and GraphEBM, we extend implementations in a public domain adaptation library DIG (Liu et al., 2021). We extend implementations of GVAE [7], Graphite [8] from codes uploaded by the original authors.

For our Computation Graph Transformer model, we use 3-layered transformers for Cora, Citeseer, Pubmed, and Amazon Computer, 4-layered transformers for Amazon Photo and MS CS, and 5-layered transformers for MS Physic, considering each graph size. For all experiments to examine the benchmark effectiveness of our model in Section 5.3, we sample $s = 5$ neighbors per node. For graph statistics shown in Section 5.5, we sample $s = 20$ neighbors per node. Our code is publicly available [9] (anonymized).

---

[5] https://github.com/google/differential-privacy/tree/main/python/dp_accounting
[6] https://github.com/pytorch/opacus
[7] https://github.com/tkipf/gae
[8] https://github.com/ermongroup/graphite
[9] https://www.dropbox.com/sh/e2ukf2djimjs4ud/AACgn0oZ0oWl0N2jILK_JEy3a?dl=0

