# OpenReview forum: "Scalable and Privacy-enhanced Graph Generative Model for Graph Neural Networks"
_ICLR.cc/2023/Conference — Submitted to ICLR 2023_

### Official Review · Reviewer_q2ks · 2022-10-20

**Confidence:** 4
**Correctness:** 3
**Technical Novelty And Significance:** 2
**Empirical Novelty And Significance:** 2
**Recommendation:** 3

**Clarity, Quality, Novelty And Reproducibility:**

# Clarity:
- The paper is written well and its ideas are easy to follow

# Quality:
- I have major concerns about the quality of the generation procedure, as it is based on a lossy encoding which loses key structural information.

# Novelty:
- The work is not particularly novel. The CGT model is an incremental change on XLNet, and the privacy contributions are primarily imported from existing literature and are a direct result of applying clustering algorithms prior to using CGT.

# Reproducibility:
- I have no concerns about the reproducibility of the reported empirical results.

**Strength And Weaknesses:**

# Strengths:
- The approach is scalable and thus can apply to large graph structures, which is not common in the graph generation domain.
- The use of label information during generation is a good inductive bias and a useful property.
- The model is simple to understand.
# Weaknesses:
-  I have major concerns about the soundness of reducing graph structures to fixed sequences. In particular, the duplicate encoding scheme is problematic, as it can lead to otherwise structurally different nodes obtaining identical computational graphs! For example, the computation graph of a node A in a triangle A -> B -> C  and is identical to that of the latter A in a line graph with identical A-featured nodes at extremes, i.e., A -> B - > C -> A (both A nodes distinct). As a result, the model is losing information in the generation process by default, and this stands regardless of the dimensionality of features used. The authors argue that the features can encode the lost structure, and discuss this near the end of the paper. However, this is not plausible in the general case, i.e., some information can be saved, but not all. Therefore, this generation procedure is effectively flattening the graph structure, which itself means that it cannot respect key structural properties of the input graphs and instead can only recover feature information. As a result, I have strong concerns that this approach can yield structure-preserving generated samples.
- The empirical protocol, though intuitive, does not fully prove the quality of the graph generation: Given the large space of potential target functions over computation graphs, it is possible to learn different tasks of similar difficulty, particularly as the generation itself 1) could make arbitrarily difficult features with enough randomness, and 2) the encoded structure of the computation graph loses information relative to the original graph, making it harder to learn the original function in some cases. Therefore, I recommend also using a similar experimental protocol to the literature, based on visualizing graphs and comparing structural properties. I understand that the baselines are small in this case, but this is essential to appraise the quality of the generated computational graphs (both label-agnostic and per label).
- The empirical results are not convincing: The gaps between synthetic and generated dataset numbers are quite substantial in Table 1. Moreover, the correlation experiment results are not surprising: It is natural that adding noise reduces performance irrespective of model choices, so this correlation in my opinion is to be expected. I feel the same about sampling neighborhood and its effect on performance.
- The discussion on privacy in this work is very limited and makes no meaningful contribution: It simply uses a clustering algorithm prior to applying the model. Therefore, I find that the privacy contribution in this paper is significantly oversold, not least in the title. I suggest that this be downplayed when revising the paper.
- Minor: Citations should be surrounded by brackets throughout the paper.

**Summary Of The Paper:**

The paper proposes a computational graph-based method for graph generation, in which computational graphs for nodes, used in graph neural networks to compute node-level outputs, are the generation objective, rather than the graphs themselves. To this end, the paper first balances the node-level computational graphs, i.e., the k-ary trees (where k is the number of sampled node neighbors) defining the computation, by adding redundant neighbor nodes and creating duplicate nodes to break cycles. This scheme is called the duplicate encoding scheme in the paper. As the trees of this scheme are balanced and k is typically fixed, the size of these trees is therefore a constant. As such, the paper then proposes a modification of the XLNet transformer, which they term computational graph transformer (CGT), to apply to these sequences. In CGT, query and context attention masks are dictated by the computational graph structure for improved inductive bias, and positional encodings are introduced at every layer to distinguish the different layers. For privacy enhancement, a clustering scheme (such as k-means) is used to anonymize and group feature vector into a discrete set of features (the cluster centroids), and shown to yield improved privacy when used with CGT. Hence, the approach can be summarized as 1) A computation graph extraction, 2) duplicate encoding, 3) feature clustering and replacement, and 4) application of CGT to compute outputs. For generation, the reverse applies: CGT is used to infer a combination of discrete feature vectors, which can then be mapped back to a duplicate encoding computational graph.

Empirically, this method is verified on a variety of benchmarks through an experimental setup based on performance similarity. In particular, the protocol looks to generate analogous computational graphs to the original inputs using the approach, and seeks to compare the performance of a variety of models on the original and generated computational graphs. The generation is then perceived to be high-quality if the model performances are similar across both original and generated graphs. In the paper, the authors report that this is the case. Finally, the authors consider manipulations to the input graph, and apply the same protocol to these situations, showing correlations between performance changes as well as similar performance values.

**Summary Of The Review:**

Overall, I understand the paper's motivation to flatten of the graph structure into a balanced tree, so as to produce a fixed-size structure and use more established and scalable generation procedures. This flattening makes the paper's approach very efficient, but in turn loses significant structural information, and practically discards several key complexities that are inherent to graph-based tasks. In fact, it is easy to come up with scenarios where this generation procedure will ambiguate clearly distinct use cases. Moreover, I am not convinced by the empirical evaluation protocol and results. Therefore, I lean towards rejecting the paper as it stands. However, I am happy to discuss my concerns with the authors, and am open to change my verdict should they offer compelling arguments addressing my concerns.

---

> ### Author Response · Authors · 2022-11-19
> **Response to R4 (1/2)**
>
> Thank you for your thoughtful questions. Before addressing your concerns one by one below, we want to emphasize that computation graph sampling is part of GNN models, not our method. Most of your concerns about information loss are about the expressivity limits of the current GNN designs, not part of our model. Discussing how information is lost in the current computation graph-based GNN design is very important; however, our paper addresses how to generate graphs for benchmarking the existing GNN models, not proposing a new GNN model.
>
> **Q1.** *“Duplicate encoding discards several key complexities that are inherent to graph-based tasks; Structurally different nodes obtain identical computational graphs.”*
>
> **RE.** We believe you might be confusing computation graph sampling with our proposed duplicate encoding scheme. Computation graph sampling is the most widely used subgraph sampling method for GNNs to deal with large-scale graphs. Computation graph sampling was initially proposed by GraphSage [1] and implemented as the default subgraph sampling method in popular GNN libraries (e.g., pyG [2], TF-GNN [3], DGL [4]). What our proposed *duplicate encoding* is doing is modifying these computation graphs (which are already sampled by GNNs) to fix their adjacency matrix without losing any structural/feature information. All discussions you raised here are about the expressive power of computation graph-based GNN models, which is out of the scope of our paper. We note that it was proved that computation graph-based GNN models are equal to the 1-Weisfeiler-Lehman (WL) graph isomorphism test [5]. In terms of duplicate encoding, we showed the key structural properties on the original computation graphs (# zero feature vectors and # duplicate feature vectors) are well preserved on the generated computation graphs in Figure 4 in Section 5.5.
>
> [1] Matthias Fey and Jan Eric Lenssen. Fast graph representation learning with pytorch geometric. arXiv preprint arXiv:1903.02428, 2019.
>
> [2] Oleksandr Ferludin, et al. Tf-gnn: Graph neural networks in tensorflow. arXiv preprint arXiv:2207.03522, 2022.
>
> [3] Minjie Wang, et al. Deep graph library: A graph-centric, highly-performant package for graph neural networks. arXiv preprint arXiv:1909.01315, 2019.
>
> [4] William L Hamilton, et al. Inductive representation learning on large graphs. NeurIPS 2017.
>
> [5] K. Xu, et al. How powerful are graph neural networks? ICLR 2019
>
> **Q2.** *“Recommend visualizing graphs and comparing structural properties”*
>
> **RE.** In Section 5.5, we described why we could not visualize the whole graph view because we generate a set of computation graphs for privacy purposes; nodes in the generated computation graphs do not have node ids which are essential to assemble computation graphs into one graph. Thus we could not compute the traditional structural properties (e.g., diameter, number of triangles) on our generated graphs. We instead introduced new structural properties for duplicate-encoded computation graphs and showed CGT preserves them across different graphs in Section 5.5 (Figure 4).
>
> **Q3.** *“I understand baselines are small in this case. This is essential to appraise the quality of the generated computational graphs (both label-agnostic and per-label)”*
>
> **RE.** For a better understanding of the quality of the generated computational graphs, we added the ablation study in Table 13. This shows that our careful design results in high correlations, while ablated models work poorly (e.g., 0.988 => 0.718 => 0.647 => -0.097 Pearson correlation rates as we remove label conditioning, positional embeddings, and masked attention modules from our original model one by one). You can also check the quality of the generated computation graphs gradually degrades as higher privacy costs are required in Table 12 (e.g., 0.934 => 0.916 => 0.862 => 0.030 Pearson correlation ratios as the minimum cluster size \\(k\\) increases 2 => 100 => 500 => 1000). For your convenience, we attached the summary of correlation ratios in Tables 12 and 13 below.
>
> | Table 12. Privacy-performance trade-off | Original | k=2 | k=100 | k=500 | k=1000 |
> |---|---|---|---|---|---|
> | Pearson | 1.000 | 0.934 | 0.916 | 0.862 | 0.030 |
> | Spearman | 1.000 | 0.935 | 0.947 | 0.812 | 0.018 |
>
> | Table 13. Ablation study | Original | w/o Label | w/o Position | w/o Attention | Ours |
> |---|---|---|---|---|---|
> | Pearson | 1.000 | 0.718 | 0.647 | -0.097 | 0.988 |
> | Spearman | 1.000 | 0.799 | 0.610 | 0.149 | 0.911 |
>
> **Q4.** *“The gaps between synthetic and generated dataset numbers are quite substantial”*
>
> **RE.** Our quantization phase (intentionally) loses feature information by mapping different node features into shared cluster ids to achieve the k-anonymity privacy guarantee. This leads to the gaps between GNN performance on original and generated graphs. However, we preserve the performance trends of different GNN models across diverse variations of a given graph.

---

> > ### Author Response · Authors · 2022-11-19
> > **Response to R4 (2/2)**
> >
> > **Q5.** *“The correlation experiment results are not surprising; It is natural that adding noise reduces performance irrespective of model choice; the Same as sampling neighborhood and its effect on performance”*
> >
> > **RE.** We do not add noise to the generated graphs; we added noise to the original graphs, and CGT is required to learn these different ratios of noisy edges on the source graphs and reproduce these noises on the synthetic graphs. Performance degrading on the original graphs via adding noises is natural, as you mentioned, however, preserving the performance degradation on synthetic graphs is not trivial. Similarly, learning distribution shifts between training and test phases is not trivial as graph generative models are required to learn distributions on training and test phases individually while preserving the distribution distance between them.
> >
> > **Q6.** *“No meaningful contribution to privacy.”*
> >
> > **RE.** Our core contribution is a newly defined graph generation problem where privacy for the original graphs is considered a requirement. Please note that none of the previous graph generative models consider privacy in their problem. Our proposed method, CGT, satisfies this privacy requirement by carefully designing the model, thus adopting off-the-shelf privacy modules while accomplishing several other requirements comprehensively (scalability and benchmark effectiveness). Our main goal is to make it possible to introduce privacy technology in the graph generation domain, paving the way in this direction.
> >
> > **Q7.** *“There are not many novel contributions in the paper; The CGT model is an incremental change on XLNet, and the privacy contributions are primarily imported from existing literature.”*
> >
> > **RE.** Using graph generative models to generate benchmark graphs for GNN research is a novel approach (Section 1). Defining three requirements for this benchmark graph generation is also a novel contribution (Section 1). Instead of focusing on the whole graph generation, we learn distributions of computation graphs to cope with the scalability issues in graph generation, which is also a novel approach (Section 3.1). Duplicate encoding for computation graphs to fix their adjacency matrices is also first proposed by us (Section 3.3). Quantization (Section 3.4) and incremental change on XLNet (Section 4.1) are just one part of our proposed ideas.
> >
> > **Q8.** *“Minor: Citations should be surrounded by brackets throughout the paper”*
> >
> > **RE.** We updated them.
> >
> > **Reproducibility:** We added an (anonymized) URL to our source code in Appendix A.10.

---

> > > ### Comment · Reviewer_q2ks · 2022-11-20
> > > **Reviewer Response**
> > >
> > > I thank the authors for their response, and appreciate the additional experimental results and clarifications.
> > >
> > > First, let me just clarify that I am aware of and fully agree with the obvious limitations of computational graphs, and that I fully accept that they are not perfect means to process graph structures. However, the reliance on the computational graph for generation is what I find problematic about this approach, and this is not a misunderstanding of the paper's scope. To be more precise, my point is that this approach does not allow us to generate graphs, i.e., sets of nodes and edges, in the standard sense, but instead relies on the computational graph, and thus **inherits its limitations** for the generative setting, causing a loss of information relative to the input graphs available in graph-structured data. Going back to my line versus triangle example, my point is that generation based on computational graphs cannot recover all tasks. For instance, one cannot learn to generate a triangle detection dataset (a task that, for example, is prevalent in social networks), as 1-WL is incapable of solving this problem, implying that the computational graphs cannot solve the task. Hence, this approach is a priori limited to 1-WL-friendly generation tasks, and thus is **tied to a modelling limitation**, whereas typical benchmarks and datasets for graph evaluation are not, and most importantly, **should not be**.  Hence, my point is the following: This generation procedure introduces a loss of generality which is not standard (see, e.g., graph generation methods like GRAN [1], which do not do this), and which is harmful in the long run, as this would eventually simplify baselines to become within the scope of 1-WL, whereas the pressing research questions in the field is in the opposite direction, namely how to overcome the 1-WL limitation in our current models. I hope this clarifies the point. In any case, I appreciate that the authors made this decision for the purposes of scalability: I just think that this decision ultimately costs more from a scientific perspective (producing general graph-level tasks with structures that are regularly sought in practice, like triangle detection) than is gained through scalability.
> > >
> > > - Q2, Q4, Q5: I understand. Thanks for the clarification!
> > > - Q3: Many thanks for the experiments. I appreciate the effort, and this definitely supports your argument.
> > > - Q6: I am afraid I am not convinced. The technical innovation for using this approach for privacy is unfortunately not sufficient, as this can be easily done anyhow. I still think that this aspect of the work should not be referred to as a main contribution.
> > > - Q7: See first point above on my concerns about this technique.
> > >
> > > As it stands, I will keep my score.
> > >
> > > [1] Liao et al. Efficient Graph Generation with Graph Recurrent Attention Networks, NeurIPS 2019.

---

> > > > ### Author Response · Authors · 2022-11-21
> > > > **Response to R4**
> > > >
> > > > We strongly disagree with this point, as our focus is limited to the scope of **GNN evaluation**.
> > > >
> > > > 1. Our paper is motivated by a lack of benchmark graphs for GNN evaluation (as mentioned in our title and introduction). We exploit graph generative models to solve this problem.
> > > >
> > > > 2. The limitations of our graph generative model are equivalent to the limitations of the existing GNN design, meaning GNN models won't be affected by our limitations.
> > > >
> > > > 3. No existing graph generative model is suitable to generate graphs for GNN evaluation. The GRAN paper you mention illustrates this point well. They generate a graph topology for a general purpose. However, we need *more than a graph topology for GNN evaluation* --- we need to generate a triad of (graph topology, node attributes, and node labels). Only our proposed method can comprehensively generate this graph input triad while satisfying *scalability* and *privacy* requirements.
> > > >
> > > > We believe our work lays the groundwork for future research in the area, which could address your concerns. However, we are confident in the merit of our work.

---

### Official Review · Reviewer_qqDr · 2022-10-21

**Confidence:** 3
**Correctness:** 3
**Technical Novelty And Significance:** 3
**Empirical Novelty And Significance:** 3
**Recommendation:** 6

**Clarity, Quality, Novelty And Reproducibility:**

Clarity: This paper is readable, giving me a feel of following a step-by-step hands-on tutorial. One mistake might be the statement that "higher $k$ and smaller $\epsilon$ hinder the generative model...". I agree with that smaller $\epsilon$ leads to stronger privacy, but higher $k$ should preserve more information in the original node features.

Quality: The design of this method has comprehensively considered the introduced three aspects of concerns in graph generation problem. The duplicate scheme seems simple yet effective in reducing the search space, which is a key technical contribution in my opinion. The modeling is natural and tailored to graph generation problem. It is great to see the authors focus on evaluating their method from those three aspects while remembering to discuss the traditional graph statistics.

Novelty: The definition of the problem is novel to me, which happens to match the demand raised from my real-world applications. The duplicate scheme seems to be novel. Those minor tricks in modeling are straightforward, but I treasure the comprehensiveness of the authors' thinking.

Reproducibility: All the datasets, baselines, and implementations are publicly available.

**Strength And Weaknesses:**

Strengths:
1. The paper is well-written.
2. The re-defined problem with the three aspects of concerns appropriately reflect the latest demands in practice, which must be a contribution to the community. I am willing to seeing more and more works solving this problem.
3. The duplicate scheme is simple yet effective, in my opinion. It transforms the problem in a way that drastically reduces the search space.
4. The empirical studies are comprehensive.

Weaknesses:
1. How to sample node features from the cluster id and how to sample computation graphs from the learned transformer should be elaborated.
2. The privacy concern is addressed from only the attributive perspective. As for the structural perspective, there is a lack of discussion. In my opinion, the discretization of node features also helps this method reduce the risk of composing the original graph from a collection of sampled computation graphs.

**Summary Of The Paper:**

This paper re-define the problem of graph generation with concerns from three aspects: scalability, benchmarking, and privacy-preservation. In response, the authors proposes an interesting minibatch-based problem formulation and accordingly apply transformer to encode the computation graphs. Many tailored details about modeling are discussed and addressed. The proposed method is extensively evaluated from those three aspects, showing superiority over existing methods.

**Summary Of The Review:**

I have went through the whole main paper and found it is interesting and technically sound. According to my experience in federated graph learning, I think the problem re-defined and somewhat resolved in this paper is important. Thus, I tend to accept this paper.

---

> ### Author Response · Authors · 2022-11-19
> **Response to R3**
>
> We appreciate your acknowledgment of our contribution to the community by resolving the newly proposed practical graph generation problem. We answer your questions below.
>
> **Q1.** *“How does CGT sample node feature from the cluster id?”*
>
> **RE.** As we mentioned in the last three sentences of Section 3.5, we de-quantize cluster ids (generated by CGT) back into the feature vector space by replacing them with the mean of feature vectors belonging to the cluster.
>
> **Q2.** *“How does CGT sample computation graphs from the learned transformer?”*
>
> **RE.** As we mentioned in the last two sentences of Section 3.5, we regenerate a computation graph from each sequence of feature vectors (which were de-quantized from generated cluster ids) with the adjacency matrix fixed by the duplicate encoding scheme. You can check the process in Figure 2.
>
> **Q3.** *“The privacy concern is addressed from only the attributive perspective. In my opinion, the discretization of node features also helps this method reduce the risk of composing the original graph from a collection of sampled computation graphs.”*
>
> **RE.** In Section 5.5, we mentioned that CGT generates a set of computation graphs without any node ids. In other words, attackers cannot merge the generated computation graphs to restore the original graph and re-identify node information. Please note that we also showed the *k-anonymity for edge distributions* in Claim 1 (proof can be found in Appendix A.1).
>
> **Q4.** *“Higher \\(k\\) should preserve more information in the original node features”*
>
> **RE.** In our paper, \\(k\\) denotes the minimum cluster size (i.e., the minimum number of nodes collected into one cluster). When the cluster sizes increase with higher \\(k\\), there will be fewer clusters given the fixed number of nodes in a graph. Then the number of distinct feature vectors in the generated graphs, which corresponds to the number of clusters, will decrease too. Finally, we will lose more feature information by having fewer distinct feature vectors.

---

> > ### Comment · Reviewer_qqDr · 2022-11-26
> > **Discussion**
> >
> > Thanks for your explanation!

---

### Official Review · Reviewer_Dpqx · 2022-11-03

**Confidence:** 4
**Correctness:** 3
**Technical Novelty And Significance:** 3
**Empirical Novelty And Significance:** 3
**Recommendation:** 6

**Clarity, Quality, Novelty And Reproducibility:**

This paper is well written with clear motivation and technical details. Novel techniques are proposed to deal with a newly formulated problem. The reproducibility should also be fine given the detailed description.


**Strength And Weaknesses:**

**Strength**
- The proposed graph generative model fills the gap in existing works to handle privacy and node attributes/labels on large graphs
- The paper proposes a clear and novel formulation for such a graph generation problem
- Designs (e.g. computation graph minibatches, duplicate encoding scheme, quantization) are well motivated and technically sound

**Weaknesses**
- Important details (e.g. how to align cluster assignments across batches, how to do de-quantization etc) are not introduced
- More experiments are needed to further verify the usefulness of generated graphs


**Summary Of The Paper:**

This paper introduces a scalable and privacy-enhanced graph generative model to learn and reproduce the distribution of real-world graphs with node attributes/labels. The proposed model satisfies benchmark effectiveness, scalability and privacy guarantee.


**Summary Of The Review:**

This work in general provides clear problem formulation and interesting ideas to transform the problem of graph generation into sequence generation. The design and evaluation would be more convincing if the following questions can be answered.

**Method**

1. What could be the advantages/disadvantages for using the computation graph, compared with a random walk based method to sample the graph? The reason for asking this question is that the cost-efficient version of CGT can be directly achieved by random walks. Meanwhile, I am curious about how sensitive this proposed method is to the sample size of neighbors $s$?

2. The cluster assignments across different computation graphs or minibatches might be different as the quantization is not conducted over all the nodes. For example, two nodes with similar features might be assigned with different cluster ids (thus are different tokens) as they are quantized separately in two batches. How to handle such a misalignment of cluster assignment?

3. The method section cuts off after introducing CGT. However, it is also important to explain how the de-quantization is conducted to finally generate computation graphs.

4. With label conditioning, can this method handle heterophilic graphs?

**Experiment**

5. Is the size of original and generated graphs comparable in the evaluations?

6. The goal of generated graphs is to train a useful GNN model that can be applied to real-world data. Therefore a more important setting should be tested: compare the performance of two GNNs on the same real-world test data, and these two GNNs are trained on the original or generated graphs respectively.

7. To verify the effectiveness of the generated graphs, it would be better if the proposed method is compared with some existing graph generative models on small datasets, such as molecular datasets.

---

> ### Author Response · Authors · 2022-11-19
> **Response to R2 (1/2)**
>
> Thank you for acknowledging the novelty of our work and the necessity of our newly defined graph generation problem. We answer your questions in addition to highlighting where we described the dequantization phase in our paper.
>
> **Q1.** *“The cluster assignments across different computation graphs or mini-batches might be different as the quantization is not conducted over all the nodes”*
>
> **RE.** Clustering happens across all the node features in the graph. In other words, all nodes across all computation graphs share the cluster assignments.
>
> **Q2.** *“How is the de-quantization conducted to generate computation graphs?”*
>
> **RE.** As we mentioned in the last three sentences of Section 3.5, we de-quantize cluster ids (generated by CGT) back into the feature vector space by replacing them with the mean of all feature vectors belonging to the cluster. Finally, we regenerate a computation graph from each sequence of feature vectors with the adjacency matrix fixed by the duplicate encoding scheme.
>
> **Q3.** *“What could be the advantages/disadvantages of using the computation graph compared with a random walk-based method to sample the graph? The cost-efficient version of CGT can be directly achieved by random walks.”*
>
> **RE.** This is a nice analysis that the cost-efficient version of CGT can be directly achieved by random walks. The only difference between computation graph-based and random walk-based modeling would be the complexities of transformer computation, as we described in Claim 3.  With the random walk-based modeling (i.e., cost-efficient version), the transformer learns shorter sequences (\\(L\\)) with the larger training set (\\(s^Ln\\)), while the computation graph-based modeling consumes longer sequences (\\(s^{2L}\\)) with the smaller training set (\\(n\\)). Random walk-based modeling seems to explore wider neighborhoods with its randomness; however, we sample a new computation graph for each node for every epoch to explore the wider neighborhood.
>
> **Q4.** *“How sensitive is this proposed method to the sample size of neighbors?”*
>
> **RE.** As the sampling size of neighbors increases, the computation cost will increase, while the generation performance can be maintained by increasing the transformer size. As shown in Claim 3, as the sample size \\(s\\) increases, longer sequences (or more training data for the cost-efficient version) will be fed into the transformer. Then, to learn distributions of longer sequences (or more sequences in the cost-efficient version), we need to increase the capacity of the transformer by adding more layers or more attention heads. In terms of GNN performance, the higher neighbor sample size for computation graphs could lead to higher accuracy on both original and synthetic graphs
>
> **Q5.** *“With label conditioning, can this method handle heterophilic graphs?”*
>
> **RE.** Yes. Our label conditioning is specifically designed for the feature-heterophilic setting. In feature-heterophilic graphs, some nodes have features that are not well-aligned with their labels, while their neighboring nodes have well-aligned features with the labels. Using label conditioning, we can generate these heterophilic patterns of features on each computation graph. Without label conditioning, correlation ratios plummet from \\(0.988\\) to \\(0.718\\), as presented in the ablation study (Table 13, Appendix A6)
>
> **Q6.** *“Is the size of the original and generated graphs comparable in the evaluations?”*
>
> **RE.** One generation step in CGT outputs one computation graph. In other words, you can decide the size of generated graphs by deciding how many generation steps you will take as follows: \\(N\\) generation steps = \\(N\\) computation graphs = \\(N\\) nodes. In our experiments, we generate the same number of nodes on generated graphs as the original graphs.
>
> **Q7.** *“Compare the performance of two GNNs on the same real-world test data, and these two GNNs are trained on the original or generated graphs, respectively”*
>
> **RE.** Great suggestions. We will add the suggested scenario to the benchmark effectiveness part. For your information, the scenario we have focused on in the paper is not a lack of training set, but the absence of the whole graphs for GNN research due to the privacy issue: the research team runs m GNN models on the generated graphs (i.e., both training and test sets from the generated graphs), picks the best one (or find trends in the results), and shares it with the product team; thus the product team does not need to start the GNN evaluations from scratch.

---

> > ### Author Response · Authors · 2022-11-19
> > **Response to R2 (2/2)**
> >
> > **Q8.** *“To verify the effectiveness of the generated graphs, it would be better if the proposed method is compared with some existing graph generative models on small datasets, such as molecular datasets”*
> >
> > **RE.** We target a single large-scale graph, such as e-commerce and social networks. As molecular datasets are given as a set of multiple small-scale graphs with only a few node/edge attributes, we thought the molecular graphs are out of our target graph sets. For a better understanding of the quality of the generated computational graphs, we added the ablation study in Table 13. This shows that our careful design results in high correlations, while ablated models work poorly (e.g., 0.988 => 0.718 => 0.647 => -0.097 Pearson correlation rates as we remove label conditioning, positional embeddings, and masked attention modules from our original model one by one). You can also check the quality of the generated computation graphs gradually degrades as higher privacy costs are required in Table 12 (e.g., 0.934 => 0.916 => 0.862 => 0.030 Pearson correlation ratios as the minimum cluster size \\(k\\) increases 2 => 100 => 500 => 1000). For your convenience, we attached the summary of correlation ratios in Tables 12 and 13 below.
> >
> >
> > | Table 12. Privacy-performance trade-off | Original | k=2 | k=100 | k=500 | k=1000 |
> > |---|---|---|---|---|---|
> > | Pearson | 1.000 | 0.934 | 0.916 | 0.862 | 0.030 |
> > | Spearman | 1.000 | 0.935 | 0.947 | 0.812 | 0.018 |
> >
> > | Table 13. Ablation study | Original | w/o Label | w/o Position | w/o Attention | Ours |
> > |---|---|---|---|---|---|
> > | Pearson | 1.000 | 0.718 | 0.647 | -0.097 | 0.988 |
> > | Spearman | 1.000 | 0.799 | 0.610 | 0.149 | 0.911 |
> >
> > **Reproducibility:** We added an (anonymized) URL to our source code in Appendix A.10.

---

### Official Review · Reviewer_jEpM · 2022-11-04

**Confidence:** 4
**Clarity, Quality, Novelty And Reproducibility:** 1. I think the clarity of sections 3-…
**Correctness:** 2
**Technical Novelty And Significance:** 3
**Empirical Novelty And Significance:** 3
**Recommendation:** 3

**Strength And Weaknesses:**

#### Strengths
1. To the best of my knowledge, I think the computation graph transformation method proposed by this paper is a novel approach to enhance privacy (more specifically, to achieve the k-anonymity privacy guarantee) on the fly and efficiently.
2. The authors considered using Pearson & Spearman ranking correlations to measure how the performance order of GNNs is preserved on the synthetic graph.

#### Weaknesses
1. One major issue is that the "benchmark effectiveness," which concerns how the synthetic graph preserves the performance ranking of GNNs, is definitely the most important metric of the proposed methodology. However, how the proposed CGT method achieves this goal is still hard to understand given the following reasons (please correct me if I am wrong): (1) The descriptions in sections 3 and 4 are mostly focused on the design and implementation of the CGT algorithm. However, it is hard to find the intuition or theoretical guarantee that ensures the benchmark effectiveness of CGT. (2) Tables 1 and 2 measure the Spearman rank correlations on multiple datasets; however, the number of GNN architectures considered (which are only GCN, GIN, SGC, GAT according to appendix A.7.1) is too small to draw a meaningful and conclusive result. Many more architectures are needed to add to the pool. Another issue is that the authors should also consider simpler and more meaningful graph generative model baselines in Table 5 in appendix A.2 (I also suggest moving Table 5 to the main paper). Otherwise, it is hard to understand the correlation metrics of CGT.
2. In terms of the scalability of the proposed CGT method, (1) the theoretical complexities claimed in Claim 3 still suffer from the exponential dependence on the number of layers, similar to a standard neighbor sampling scheme, and this hinders the application to deeper GNNs. (2) I only see a few experimental results on the scalability comparison (e.g., Table 5 in appendix A.2 only shows some methods out of memory). Some more measures like the computational time and memory usage of CGT and some simpler yet meaningful graph generative model baselines are encouraged to be added.

**Summary Of The Paper:**

This paper aims to tackle the privacy issue when collecting datasets from online databases by introducing a novel graph generative model, which is called Computation Graph Transformer (CGT), that can learn the distribution of real-world graphs in a privacy-enhanced manner. The proposed algorithm can generate synthetic
substitutes of large-scale real-world graphs, which can be used to benchmark GNN models. First, CGT operates on computational graphs of mini-batches rather than the whole graph, avoiding scalability issues. Second, a novel duplicate encoding learns a single, dense feature matrix instead of learning the adjacencies and features simultaneously


**Summary Of The Review:**

Overall I recommend the rejection of the current manuscript. I agree with the novelty and the importance of the problem considered: to enhance the privacy of the real-world, large-scale graphs collected from online databases. But due to the current insufficient theoretical analysis and experiments that justify CGT solves the benchmark effectiveness and scalability goals as the paper claimed, I cannot recommend acceptance now. Either some theoretical guarantees on how the benchmark effectiveness is achieved, or a much larger GNN architecture pool to measure the rank correlation is needed.

---

> ### Author Response · Authors · 2022-11-19
> **Response to R1 (1/2)**
>
> Thank you for your thoughtful questions. We addressed your main concerns in benchmark effectiveness and scalability as follows.
>
> **Q1. Benchmark effectiveness**
>
> **Q 1.1.** *"It is hard to find the intuition or theoretical guarantee that ensures the benchmark effectiveness of CGT."*
>
> **RE.** You are correct that CGT does not directly guarantee, nor directly optimize for, any metric of benchmark effectiveness. Any GNN benchmark effectiveness metric must involve the correlation in performance on original and synthetic graphs for a finite set of GNN models. Although directly optimizing for a benchmark effectiveness metric is an interesting avenue for future work, we chose not to pursue it because it could easily lead to overfitting to the particular GNN models used to compute the metric. Instead, our proposed graph generative model CGT is optimized/tuned/trained to generate synthetic graphs that reproduce the local properties (i.e., computation graphs) of the target graph by optimizing an *auto-regressive loss* (Equation in Section 4.1). The intuition for why this should lead to benchmark effectiveness is that the original graph itself would trivially be the *most* effective benchmark substitute, and thus optimizing for the generation of synthetic graphs close to this original graph should lead to benchmark effectiveness.
>
> **Q 1.2.** *"The number of GNN architectures considered (which are only GCN, GIN, SGC, GAT according to appendix A.7.1) is too small to draw a meaningful and conclusive result. Many more architectures are needed to add to the pool."*
>
> **RE.** We run 336 (7 graph datasets x 12 versions per dataset x 4 GNN models per version) different GNN performance comparisons between original and generated graphs on node classification tasks and 40 (5 graph datasets x 4 GNN models x 2 link predictors) different comparisons on link prediction tasks using **9 different GNN models** to examine the benchmark effectiveness of our model. You can find these results in Tables 6, 7, 8, 9, and 10 in Appendix A.3.
>
> We evaluate whether our proposed model can effectively capture 1) performance differences between different GNN models on the same graph and 2) performance differences of the same GNN model across different versions of a given graph. We first choose 9 target GNN models: 4 of them propose different aggregation strategies (*GCN, GIN, SCG, GAT*), another 4 propose different neighborhood sampling methods (*AS-GCN, GraphSage, BS-GCN, PASS*), and another subset (*GAT, SCG, GraphSage, PPNP*) is used to evaluate robustness to distribution shift as proposed in [1]. As these 9 GNN models put emphasis on different aspects of GNN modeling, we split their evaluation across 4 different scenarios: 1) how each aggregation strategy is affected by noise, 2) how each sampling strategy is affected by noise, 3) how each sampling strategy is affected by the sampling number, and 4) how each GNN model is affected by distribution shift. For each scenario, we construct 3 variations of the graph: 1) different number of noisy edges for aggregation, 2) different number of noisy edges for sampling, 3) different sampling number, and 4) different amount of distribution shift, respectively. In the final version of the paper, we will add additional GNN models to experiments, but we believe the current number of experiments is large enough to draw meaningful conclusions.
>
> We also added Pearson and Spearman correlation ratios across all 9 different GNN models on all 7 datasets in Table 11 in Appendix A.3 (without any variations on graphs). We attached the summarized correlation ratios below. Our proposed CGT successfully preserves the performance trends among 9 different GNN models (*GCN, GIN, SCG, GAT, AS-GCN, GraphSage, BS-GCN, PASS, PPNP*) across all datasets with up to 0.964 Pearson correlation ratios.
>
> |           | Across 9 GNN models |          |
> |-----------|--------------------|----------|
> | Dataset   | Pearson             | Spearman |
> | Cora      | 0.939               | 0.868    |
> | Citeseer  | 0.948               | 0.743    |
> | Pubmed    | 0.962               | 0.868    |
> | AmazonC   | 0.952               | 0.920    |
> | AmazonP   | 0.899               | 0.786    |
> | MS CS     | 0.964               | 0.784    |
> | MS Physic | 0.947               | 0.776    |
>
> [1] Qi Zhu, et al. Shift-robust gnns: Overcoming the limitations of localized graph training data. Neurips, 2021.

---

> > ### Author Response · Authors · 2022-11-19
> > **Response to R1 (2/2)**
> >
> > **Q 1.3.** *"Authors should also consider simpler and more meaningful graph generative model baselines in Table 5 in appendix A.2 (I also suggest moving Table 5 to the main paper). Otherwise, it is hard to understand the correlation metrics of CGT."*
> >
> > **RE.** Please note that, in this paper, we introduce a novel problem in graph generation — namely, generating large-scale graphs with node features and labels. While we fully agree that meaningful baselines are important, we found that, unfortunately, five of the most popular/state-of-the-art graph generative models (*GVAE, Graphite, GraphAF, GraphDF,* and *GraphEBM*) either do not scale or do not have good benchmark effectiveness in our novel setting. This is somewhat expected, since these models were developed to solve a different graph generation problem -- namely, generating many small-scale graph structures from a target small-scale graph distribution. We welcome any additional baseline model suggestions from the reviewer.
> >
> > For a better understanding of the quality of the generated computational graphs, we added the ablation study in Table 13. This shows that our careful design results in high correlations, while ablated models work poorly (e.g., 0.988 => 0.718 => 0.647 => -0.097 Pearson correlation rates as we remove label conditioning, positional embeddings, and masked attention modules from our original model). You can also check the quality of the generated computation graphs gradually degrades as higher privacy costs are required in Table 12 (e.g., 0.934 => 0.916 => 0.862 => 0.030 Pearson correlation rates as the minimum cluster size \\(k\\) increases 2 => 100 => 500 => 1000). For your convenience, we attached the summary of correlation ratios in Tables 12 and 13 below.
> >
> > | Table 12. Privacy-performance trade-off | Original | k=2 | k=100 | k=500 | k=1000 |
> > |---|---|---|---|---|---|
> > | Pearson | 1.000 | 0.934 | 0.916 | 0.862 | 0.030 |
> > | Spearman | 1.000 | 0.935 | 0.947 | 0.812 | 0.018 |
> >
> > | Table 13. Ablation study | Original | w/o Label | w/o Position | w/o Attention | Ours |
> > |---|---|---|---|---|---|
> > | Pearson | 1.000 | 0.718 | 0.647 | -0.097 | 0.988 |
> > | Spearman | 1.000 | 0.799 | 0.610 | 0.149 | 0.911 |
> >
> >
> > **Q2. Scalability of the proposed CGT method**
> >
> > **Q2.1.** *“The theoretical complexities claimed in Claim 3 still suffer from the exponential dependence on the number of layers, similar to a standard neighbor sampling scheme, and this hinders the application to deeper GNNs.”*
> >
> > **RE.** The exponential dependence on the number of layers O(\\(s^L\\)) corresponds to the sampled computation graph size. GNN models sample computation graphs to deal with large-scale real-world graphs; thus, computation graph sizes are commonly limited (with small numbers for \\(s\\) and \\(L\\)). To deal with deeper GNNs, we can disentangle the depth of computation graphs and the depth of GNN models as proposed in [1].
> >
> > [1] Zeng, Hanqing, et al. "Decoupling the depth and scope of graph neural networks." NeurIPS 2021
> >
> > **Q2.2.** *“I only see a few experimental results on the scalability comparison (e.g., Table 5 in appendix A.2 only shows some methods out of memory). Some more measures like the computational time and memory usage of CGT and some simpler yet meaningful graph generative model baselines are encouraged to be added”*
> >
> > **RE.** Our focus is on evaluating whether CGT and baseline graph generative models can be scaled to large-scale graphs (e.g., Microsoft Physics graph with \\(\sim\\)35,000 nodes and \\(\sim\\)250,000 edges, the largest graph dataset existing graph generative models have ever been evaluated on [1]) while showing reasonable benchmark effectiveness. Table 5 successfully demonstrates this: our model is *practically runnable* at this graph scale on a 16GB memory GPU: only our model shows reasonable benchmark effectiveness (0.97~0.99 correlation rates) while all baselines either run out of GPU memory or fail to show meaningful GNN performance. In Claim 3, we also provide a theoretical analysis of why our model computation scales linearly with the size of the graph. We released our code (Appendix A.10), and the datasets we used are publicly available, allowing easy reproducibility.
> >
> > [1] Guo, Xiaojie, and Liang Zhao. "A systematic survey on deep generative models for graph generation." IEEE Transactions on Pattern Analysis and Machine Intelligence (2022).
> >
> > **Reproducibility**: We added an (anonymized) URL to our source code in Appendix A.10.

---

### Author Response · Authors · 2022-11-19
**Response to All Reviewers**

We thank all reviewers for their time and valuable comments. In this work, we propose a new graph generation problem to overcome the unavailability of real-world benchmark graphs due to their proprietary nature.

We appreciate reviewers acknowledge that:
- *“This work fills the gap in existing graph generation works to handle privacy and node attributes/labels on large graphs”* (Reviewer 2. Dpqx)
- *“The re-defined problem reflects the latest demands in practice, which must be a contribution to the community. I am willing to see more and more works solving this problem”* (Reviewer 3. qqDr)
- *“According to my experience in federated graph learning, I think the problem re-defined and somewhat resolved in this paper is important.”* (Reviewer 3. qqDr)

We believe we addressed Reviewer 1. jEpM’s concerns about benchmark effectiveness and scalability. Reviewer 4. q2ks’s concerns about how information is lost in the current (computation graph-based) GNN design, while important to discuss as a community, are out of the scope of our paper. We address how to generate graphs to benchmark *existing GNN models*, not how to improve current GNN design.

---

### Decision · Program_Chairs · 2023-01-20

**Decision:**

Reject

**Justification For Why Not Higher Score:**

The reviewers pointed to the concerns including:  insufficient theoretical analysis and experiments that justify CGT solves the benchmark effectiveness and scalability goals, questionable scalability to deeper GNNs, further evidences on the usefulness of generated graphs, loss of information due to duplicate encoding scheme, limited discussions on privacy insufficient for stated claims. The authors provided point-to-point responses that addressed some of these concerns. However, the remaining concerns left two of the reviewers’ still relatively unconvinced. Overall, there still seems to be a lack of enough enthusiasm among the reviewers after the author responses.

**Justification For Why Not Lower Score:**

N/A

**Metareview: Summary, Strengths And Weaknesses:**

This paper introduces a scalable and privacy-enhanced graph generative model, which is called Computation Graph Transformer (CGT), to learn and reproduce the distribution of real-world graphs with node attributes/labels. CGT operates on computational graphs of mini-batches rather than the whole graph, avoiding scalability issues. Duplicate encoding learns a single, dense feature matrix instead of learning the adjacency and features simultaneously. Empirically, this method is verified on a variety of benchmarks through an experimental setup based on performance similarity. The paper also discussed on scalability and privacy-preservation. The reviewers pointed to the concerns including:  insufficient theoretical analysis and experiments that justify CGT solves the benchmark effectiveness and scalability goals, questionable scalability to deeper GNNs, further evidences on the usefulness of generated graphs, loss of information due to duplicate encoding scheme, limited discussions on privacy insufficient for stated claims. The authors provided point-to-point responses that addressed some of these concerns. However, the remaining concerns left two of the reviewers’ still relatively unconvinced. Overall, there still seems to be a lack of enough enthusiasm among the reviewers after the author responses.